# Modeling Correlation between Android Permissions Based on Threat and Protection Level Using Exploratory Factor Plane Analysis

**Moses Ashawa *** and **Sarah Morris**

Digital Investigation Unit/Centre for Electronic Warfare, Information, and Cyber/Defence and Security, Defence Academy of the UK, Cranfield University, Shrivenham SN6 8LA, UK; s.i.morris@cranfield.ac.uk
* Correspondence: m.ashawa@cranfield.ac.uk; Tel.: +44-73-9489-1101

**Abstract:** The evolution of mobile technology has increased correspondingly with the number of attacks on mobile devices. Malware attack on mobile devices is one of the top security challenges the mobile community faces daily. While malware classification and detection tools are being developed to fight malware infection, hackers keep deploying different infection strategies, including permissions usage. Among mobile platforms, Android is the most targeted by malware because of its open OS and popularity. Permissions is one of the major security techniques used by Android and other mobile platforms to control device resources and enhance access control. In this study, we used the t-Distribution stochastic neighbor embedding (t-SNE) and Self-Organizing Map techniques to produce a visualization method using exploratory factor plane analysis to visualize permissions correlation in Android applications. Two categories of datasets were used for this study: the benign and malicious datasets. Dataset was obtained from Contagio, VirusShare, VirusTotal, and Androzoo repositories. A total of 12,267 malicious and 10,837 benign applications with different categories were used. We demonstrate that our method can identify the correlation between permissions and classify Android applications based on their protection and threat level. Our results show that every permission has a threat level. This signifies those permissions with the same protection level have the same threat level.

**Keywords:** cybersecurity; mobile malware; factor analysis; dangerous permission variables; protection level; Bayesian correlation; threat level

## 1. Introduction

Among the smartphone platforms, Android is the most popular. A previous study [1] showed that Android owns 74.5% of the mobile device marketplace. The popularity of Android has caused an increase in third-party application development to respond to the shift. Factors such as open-source [2,3], ease of use [4,5], and low cost [6,7] contribute significantly to its popularity and fast spread. The open-source nature of the Android OS architecture enables the unrestricted distribution of third-part applications on different distribution markets. Consequently, a wide-ranging API is provided to third-party developers. The large-scale permits Android applications to have access to the device resources and users' privacy when certain permissions are accepted by the user.

While some of the permissions are normal, Google classified certain permission variables as dangerous. Permissions variables that are classified as dangerous would likely have a higher threat level than normal permissions. To ensure that users' data are protected, Android imposes applications to request and obtain permissions before accessing sensitive data from the device. During application installation, if the user denies granting access, installation is aborted. Any permission request to access device resources and users' private information can potentially be dangerous due to the advent of millions of malicious applications using different infection and evasion techniques. Anytime an application

requests access to an Android device, the request forces the user to decide whether to accept the decision or decline the permission. While some permissions have malicious intent, most mobile users do not think of the implication of granting the permissions. Still, they are more concerned with getting the application downloaded and installed, as outlined in the research of [8]. The results obtained in the study of [9] affirm that the majority of mobile users are careless about accepting permissions than the resultant havoc such permission could cause.

While few users pay attention to the kind of permission access request by applications, malware writers deploy different ways to drip them by setting some malicious applications with default handlers to avoid multiple permission requests during runtime [10]. It then implies that once certain permission is accepted to access the device resources, others with similar features are likely to be installed in the background without presenting a further request for access [11]. The study of [12] identified that in a random sample of a given population, features of some variables have control or influence the characteristics of one or more variables at given levels of independent measurement. This signifies that acceptance of specific permissions influence the installations of the others with a certain level of correlation. It then means there is a correlation that exists between permission variables. The open issue then is how to control the behavior of permissions that malicious applications request. To achieve this, understanding the relationship between permission variables is key. Therefore, since the security of Android depends on the performance of the permission mechanism of the OS platform, it is pertinent to determine how correlated permissions are. Understanding how permissions are related will provide insights into how Android applications handle sensitive and high-risk data. Additionally, this will help in human decision making when installing applications whose source is not legitimate. The key contributions of our research are the following:

1. We present a model for visualizing the correlation between Android permissions using the t-Distribution stochastic neighbor embedding (t-SNE) and Self-Organizing Map (SOM) techniques.
2. We demonstrate results that show the relationship between a threat and protection level in Android permissions using exploratory factor plane analysis. The results show that every permission, whether normal or dangerous, has a threat level.
3. We identify that Android permissions with the same protection level have the same threat level. However, the threat level in the individual applications differ.
4. To examine Android permissions commonly requested and disseminated to classify Android applications as malicious or benign. Our results demonstrate that the proposed model can determine families of malware based on the similarities by understanding their clusters. This demonstrates that Android permissions in the same cluster have similar attributes.
5. We build on the existing work to expand Android permission request state-of-the-art by providing a comprehensive study on the current state of permissions systems.

The remainder of this paper is structured as follows. In Section 2, we provide the background on Android permission system architecture. The taxonomy of the permission flags and the protection level is described. In Section 3, we review the existing work relevant to our research. Materials and methods for our model are described in Section 4. Results and discussion are presented in Section 5. The conclusion is provided in Section 6.

## 2. Background

This section provides background to the Android permission architecture and other components. It also provides detailed background on the protection levels, permission flags, and other essential Android security components such as the API calls and message intents.

## 2.1. Android Permission Architecture and Other Components

The security architecture of Android revolved around its permission system. Based on the security architectural design of the Android platform, no application is allowed to install and function in any form if the Android OS, other applications, or users' data are affected adversely. The security architecture that regulates and enforces this is the permission system. However, due to regular updates in Android versions and code names, the permission system keeps evolving. Additionally, the range of the Android API levels increases with the increase in the number of permissions (Table 1). Each Android version released is incorporated with higher API levels and more permission features. For instance, the first Android code name (Base) released has an API level of 2 with total permissions of 73 [13,14]. However, as new code names emerged, there is an increase in the API levels and the permission features. The increase in the total number of permissions is influenced by the protection level offered by the device. The protection level is one of the permission groups defined by Android [15].

**Table 1.** Android code name release versions showing the increase in the API level and number of permissions.

| Code Name | Platform Version | API Level | Number of Permissions |
| --- | --- | --- | --- |
| Android 11 beta | 11 | 30 | 167 |
| Q | 10 | 29 | 158 |
| PI | 9 | 28 | 148 |
| Oreo | 8.0−8.1 | 26−27 | 144 |
| Nougat | 7.0−7.1 | 24−25 | 135 |
| Marshmallow | 6 | 23 | 131 |
| Lollipop | 5.0−5.1 | 21−22 | 120 |
| KitKat Watch | 4.4 W | 20 | 113 |
| KitKat | 4.4 | 19 | 112 |
| Jelly Bean | 4.1−4.3.1 | 16−18 | 104 |
| Ice Cream Sandwich | 4.0.1−4.0.4 | 14−15 | 98 |
| Honeycomb | 3.0.−3.2 | 11−13 | 95 |
| Gingerbread | 2.0−2.3.5 | 9−10 | 94 |
| Froyo | 2.2. | 8 | 87 |
| Éclair | 2.0−2.1 | 5−7 | 86 |
| Donut | 1.6 | 4 | 85 |
| Cupcake | 1.5 | 3 | 81 |
| Base | 1−1.1 | 1−2 | 73 |

## 2.2. Protection Levels and Permission Flags

The levels mean the intention of using specific permission and the resultant effects or consequences that come with using such permission. The protection level in the Android system defines the potential threat hidden in the permission. It specifies the possible measure the system should adapt when deciding whether to grant access or deny the permission requested by an application. Android permission system supports four basic types of protection levels with different permission flags (as shown in Figure 1). These include normal protection level, dangerous protection level, signature protection level, and signatureOrSytem protection level.

### 2.2.1. Normal Protection Level

Systems usually grant access to normal protection levels without needing compliance from the user. Normal permissions are sometimes misused by malware to bypass mobile device security to access device resources and other private data. Malware authors may define some functionalities as access control mechanisms to communicate with pre-defined applications to access the resources. For instance, when these permissions are excessively requested by the over-privileged applications, this poses security issues by increasing the threat level of a device.

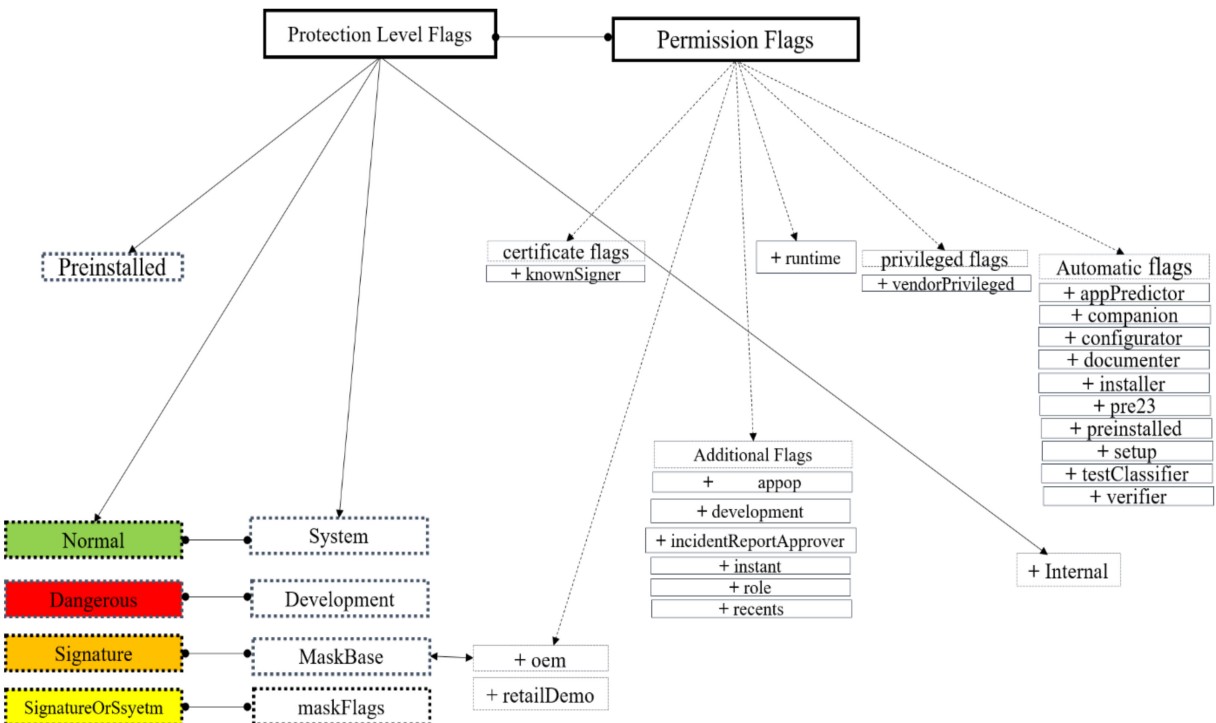

**Figure 1.** Protection level and permission-based flags.

### 2.2.2. Signature Protection Level

This protection level compares the certificate of declaring and requesting application to determine if it can be signed by the same certificate as that of the device manufacturer. Permission is granted only if the certification condition is satisfied; that is to say, signature permissions can only be granted if the same certification signs both parties. Signature permissions are only granted if the requesting application is signed by the same developer that defined the permission.

### 2.2.3. Dangerous Protection Level

Dangerous protection level grants access to the device's resources, private information of the user, device operating system, and the functioning of other applications installed on the device. Dangerous permissions are operational when users grant explicit access to them. On most occasions, dangerous permissions are presented to the user when installing third-party applications.

### 2.2.4. Signature or System Protection Level

SignatureOrSytem protection level is granted to only applications that are signed with the same system Android system mage certificate. SignatureOrSytem permissions are required where applications built by two or more vendors need specific features of the system image to be shared explicitly.

While the pre-installed applications can use permissions in all the protection level categories, only normal and dangerous permissions can be used by third-party applications. Any time permissions from the signature and SignatureOrSytem protection level are requested by third-party applications, such permissions are rejected by the OS platform. Malicious applications can leverage different permission flag constants such as pre23 [16,17] of the system protection level to automatically grant permissions to malicious applications that target API levels. Applications with permission flags granted automatically reside in the Original Equipment Manufactural (OEM) partition of the Android. Due to the connection between the OEM and the MaskBase [18] in the protection level, privileged

flags such as vendorPrivileged enables both malicious and benign applications to access any permission, especially those associated with the system image.

### 2.3. Intent Message

Malek defined an Intent message as "a trigger event for an activity or service to be performed along with the required data which supports that requested action" [19]. Intent messages, when exchanged, can be used by malicious applications to escalate privileges. The inter-component communication mechanizing [20] of Android relies chiefly on Intent messages. Intents are generally used by Android to aid data delivery via asynchronous messages. An Intent object is significant in Android permission because it holds data that the Android system uses to define recipient components needed to operate. This is significant because it enhances and eases service implementation in passing data and making it available to applications. This allows the user to initiate an action in a different application by calling a simple operation that the user would like to perform. Examples of such activities include but are not limited to taking pictures or viewing maps [21]. Intent plays a significant role as a communication mechanism in intra-application and inter-application message exchange.

### 2.4. API Calls

API stands for Application Programing Interfaces [22], whose levels are represented by integer values that uniquely identify each Android framework version. The android operating system uses APIs to reinforce third-party applications. Through APIs, entry to the device resources and features is easily made. Application programing interfaces consist of two structural components: API library and API implementation (as shown in Figure 2). API library is situated in the virtual machine (VM) of the Android application. The implementation structure of the API is the system's running process. These two API structures are packed in the Software Development Kits (SDK) of the Android platform. During the device operation process, the application library of the API calls the private interface. The private interface then invokes and initiates all the remote process calls. This helps in assigning services from the service thread during the runtime process. Critical examination of the Android API call chain can aid virtually in disclosing an application's intention. CFGs form the entire representation of the Android application. It forms a graph consisting of finite sets of nodes (N) [23] of the documented API calls and the finite set of edges (E) [24], which link successive instructions.

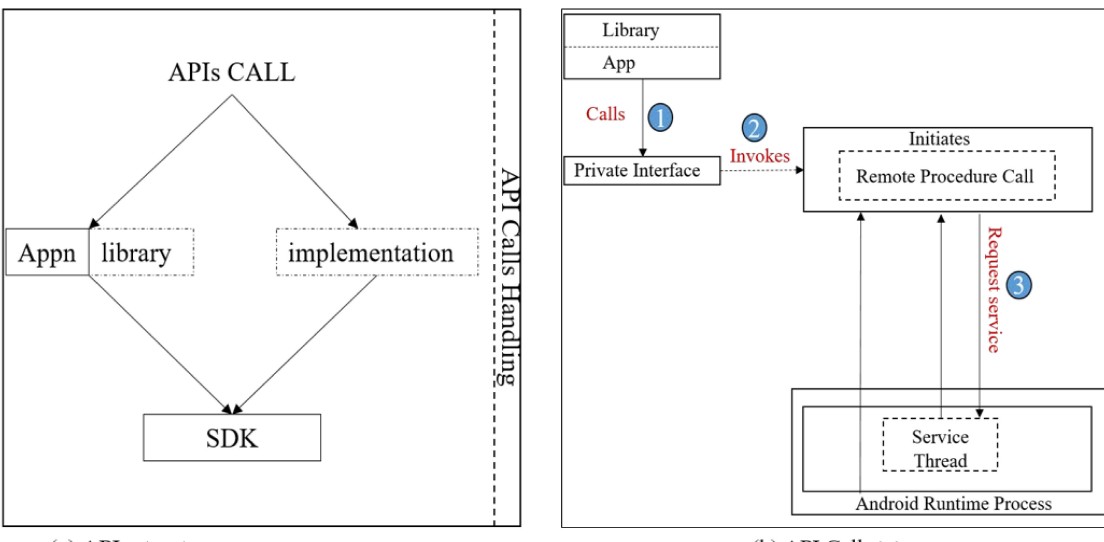

(a) APIs structure  (b) API Call states

**Figure 2.** Android OS API call architectural handling process. This figure shows the connection the APIs structure and Call states interfaces respectively.

Control Flow Graph (CFG) is a directed graph description of how a program is controlled during execution. Straight lines, nodes, and edges are some of the graphical notations used by CFG. The ability of the CFG to link the entry block helps in summarizing the control flow. When an application has malicious intent, CFG blocks the code and makes it unreachable from appending to the Android OS or existing applications. However, adversarial examples can manipulate CFGs by modifying a sample to evade detection from a classifier or a detection engine. In the same vein, malware use code level or binary to manipulate their sample byes during compilations. This manipulation enables benign applications to inject their block of bytes into malicious binary. According to [25], some code-level malware applies perturbation and then modifies the original code structure. When malware attack the CFG of an Android-based device, it causes structural modification of the code's feature space.

## 3. Related Work

The existing work for this research is classified as follows.

### 3.1. Permission-Based Detection and Feature Extraction

Several studies have been carried out on Android permission requests [26], Android user privacy [27], attacks on Android and other mobile platforms [28,29], including malware attacks [30], synchronous channels attack [31], side-channel attacks [32], runtime repackaging attacks [33], and Man-in-the-Middle attacks [34] accuracy. This section of the paper provides a brief examination of some of the related works carried out. Almomani and Al Khayer [1] discussed the Android permission systems by providing a general overview of the Android control mechanism. The study identified and grouped Android dangerous permissions based on their API levels. The research showed the significant influence of Android permissions at diverse protection levels. Zarni and Zaw [35] proposed an Android malware detection framework that collects Android applications' permissions by applying clustering algorithms. Their system monitors several permission features obtained from Android apps to determine whether the permissions are malicious or benign.

In [36], Akleylek proposed permission-based detection of Android malware using a feature selection approach based on linear regression. Using a minimum of 27 features, their research selected the most distinctive permissions to enhance the framework's performance instead of using all the permission feature vectors. In [37], Mcdonald et al. used four different algorithms to classify Android malicious applications as either malicious or benign based on their permissions manifest files. Using a dataset with a sample size of 5243, the best algorithm obtained a performance accuracy of 81.5% accuracy. The research concluded that hackers target manifest permission features as an attack vector to infect mobile devices. In [38], Mathur et al. proposed a detection framework for detecting Android malware using custom and native permissions features. Their findings show that the increasing number of Android permissions increases options to gain access and control over sensitive information and mobile devices.

In [39], Shahriar et al. proposed a framework for the detection of Android permissions that are anomalous. The detection system focused on the identification of relevant categories by applying a latent semantic index [40,41]. The result of the study shows that dangerous permissions constitute a security vulnerability. The understanding of permissions relationships could help in the detection of malicious applications. In [42], Li et al., proposed an Android malware detection tool called SigPid. Their system is based on the analysis of permission usage. Using a three-level pruning approach, their study mined the permission dataset to effectively and distinguishingly identify significant permissions in malicious and benign applications. Although their study achieved a high detection accuracy of 93.62%, only 22 out of 135 permissions were considered. Additionally, the research did not consider the protection level or threat level of those permissions.

### 3.2. Control Flow Graph and Information Gain

In [43], Arora et al. used permission pairs to identify conspicuous permissions that lead to Android malware detection. Their study compared graphs of malicious and benign applications by mining the permission pairs. However, understanding the influence of pairing permissions is significant in malware detection using permission feature attributes. The proposed detection system, however, has some limitations, one of which is that malicious applications with single or no permission cannot be detected or analyzed. Another limitation of their research is that they did not explain which pair of permissions are predominantly present in benign or malicious applications. To overcome this challenge, an understanding of the correlation between protection levels and permission flags will help in developing a robust detection tool that could detect malware with permissions run time attributes.

In [44], Khariwal and Singh used the Information Gain approach to rank intents and permission features based on their Information Gain score. Their research combined both permissions and intents attributes and then applied four different machine learning algorithms. Using the Bag-of-Words model [45–47], the extracted intents and permissions from the manifest file of the Android APK were embedded in the NLP [48,49]. Investigating the dynamic permissions using machine-learning methods to detect Android malware, dynamic permissions were extracted from a large number of public Android applications using J48, Naïve Bayes, and Random Forest algorithms in the research of Mahindru and Singh [50]. The results obtained by the algorithms were then compared based on their accuracy. However, their research did not demonstrate how correlated dynamics permissions are, and by this, false suggestions might be caused. The research of [51] compared two specific Android features: system calls and permission requests as an approach to malware detection. Their research identified some techniques which are essential in malware identification, and avoidance including USB examination and scanning and disabling mobile mode for automatic app download. However, the study did not compare machine learning algorithms to determine which could be more suitable to achieve a better and more accurate detection.

### 3.3. Bayesian Correlation, Opcode Sequence, and t-Distribution Stochastic Neighbor Embedding (t-SNE)

Bayesian correlation is one of the statistical linear mixed models that consistently produces probabilistic management of variables in a sample population. Bayes methods are used in different applications for matching functions with correlated likelihoods. The likelihoods are deployed in filtering responses used for learning training population samples [52]. Using Bayesian correlation, the research of [53] studied object localization to derive asymptotic properties of permissions with close similarities though different in their very nature. Using images as a sample population, the researchers deployed a Multiple Scale (MS) approach to determine the basis for which images with similar features can be classified using background modeling. The result of their research shows that Bayesian correlation can be effectively used for cross-correlation when using large-scale parameters for modeling relationships between variables [54]. The research of [55] used Bayesian linear regression to represent the relationship between gene particles by estimating the gene weights using correlation coefficient factors. The research used a profile dataset to model how the properties in gene A are related to gene B and vice versa.

Working on the Bayes model, the research of [56] formulated a Bayesian model using canonical analysis correlation to extract the correlation between sample data sets. In their approach, the data set decomposed into specific sets of statistical independence to help make logical inferences on other variables with similar attributes in the data set. Exploring multiple features of Android malware, the study of [57] designed a multiple feature detection framework for extraction of opcode sequence and N-Gram with symmetrical uncertainties using correlation-based models. Malware features extracted by the model were not complete in identifying some characteristics of malware. The framework was also

inadequate in demonstrating the correlation that exists among those features extracted; thus, it cannot identify emerging Android malware. Using t-distribution stochastic neighbor embedding (t-SNE) for visualization, Zhang et al. [58] proposed a method for detecting malicious adverts in addresses of web pages using lexical-based features. Even though HTTPs provide authentication on the web browsers and pages to offer data secrecy [59], the study of Barrera et al. [60] demonstrated that sophisticated malware leverage on permission-based security vulnerabilities to infect Android and web-based applications.

## 4. Materials and Methods

Various visualization techniques are used for analyses of pairwise similarities and relationships between variables in complex dataset distribution. Some of the virtualization techniques include Scatter plot [61], interpolation [62], and histogram [63], among others. These traditional virtualization techniques have limitations in facilitating virtualization of limited variables of data set at a time as highlighted when virtualizing their similarities as highlighted by Van [64]. This limits traditional virtualization techniques from being used on data sets with high dimensions and complexity like malware. To develop a model that generates correlation between permissions in the Android malware dataset, it is essential to automate the data analysis before virtualization. In this research, we used the t-Distribution stochastic neighbor embedding (t-SNE) [65] to study data low-dimensional embedding and Self-Organizing Map (SOM) [66] to present the interactive view of Android permissions and identify the relationship between them in high dimensional shape. In addition to producing inputs with high dimensions, we used in our study to manage the complication of having too many points that begin to overlap, which is the same approach used in hexagonal binning [67]. In a typical permission-based architecture, numerous permissions are at the user's disposal. In our research, the processing method focused on understanding how the Android permission model is used and demonstrating how an attacker can leverage high threat levels to escalate privileges based on the permission vulnerabilities. Our methodology allows us to gain insights on how malicious use the given permission model in practice and highlights the strengths and shortcomings of the model accordingly. We note that although the case study focuses on Android, our empirical analysis is appropriate for various other Android permission-based architectures, as long as the applications are represented as a bit string of permissions.

### 4.1. Data Set

Two categories of datasets were used for this study, the benign and malicious datasets. Dataset was obtained from Contagio [68], VirusShare [69], and Androzoo [70]. Though Drebin is an old dataset, we included part of the dataset obtained from the Impact cyber repository [71] to have a large dataset of both old and new permissions. Using VirusTotal scanner, 12,267 malicious applications were selected. A total of 10,837 benign applications with different categories were obtained from the Android PlayStore. This research is a continuation of the previous study on the Android malware permission request variables designed to classify Android permission requests based on protection and threat level. These data sets were chosen because they contain large app variants of malicious files that were collected from the official Google Play store and other official and legitimate alternative markets such as Slideme and legitimate repositories such as F-Droid.

To extract permissions from the selected Android applications, the Apktool was used to decompile the .apk file into different contents, including 'AndroidManifest.xml', 'Classes.dex', and the 'res'. All the applications permissions are contained in the 'AndroidManifest.xml', while the dex has strings and Dalvik Opcodes. The most significant features were selected using information gained by extracting the similarities between sets of permission and then calculating and scoring each permission individually. Using feature encoding, the permissions were converted into binary vectors by concatenating all the features, which is the input of the model. Details of how permission features were

extracted from the data set with other attributes and how they are converted into input vectors have been described in our previous research published in [72].

### 4.2. The t-Distribution Stochastic Neighbor Embedding

Take a data set $D$ with highly dimensional input variables and a function $f$, which determines the distance between pairs of variables. Let $D = \{y_1, y_2, \ldots, y_N\}$, and $f = e(y_i, y_j) = \|y_i, y_j\|$. To determine the K-dimensional embedding that each variable is embodied with point $\varepsilon = \{x_1, x_2, \ldots, x_N\}$, $x_i$ was expressed to belong to a set of code points expressed as $x_i \in \mathbb{R}^k$. The joint probabilities $P_{ij}$ that computes the similarity existing between permissions $y_i$ and $y_j$ is expressed as follows:

$$P_{ij} = \frac{\frac{E\left(-e\left(y_i, y_j\right)^2\right)}{2\ell_i^2}}{\frac{\sum_{k \neq 1} E\left(-e\left(y_i, y_k\right)^2\right)}{2\ell_i^2}} \tag{1}$$

$$P_{ij} = \frac{P_{j|i} + P_{i|j}}{2N} \tag{2}$$

From Equation (1) above, $E$ is the exponentiation of the $f$ function, $\ell$ represents the Gaussian kernel parameter, and $N$ is the number of variables, in our case, permissions. In our study, we defined $\ell$ in the dimension that $P_i = \mu$. The function $\mu$ is the perplexity of the conditional probability $P$, which enables optimization of the $\ell$ value. The optimal value of the Gaussian kernel differs at every point, and its value decreases in areas that the density of the data is higher in the data space, and it increases in regions of lower data density, respectively. As shown in Figure 3, the value of $\ell^2$ decreases in regions where the slope is steep. This shows how the weighting matrices can be influenced when determining the correlation between two permissions in the data set. The k-dimensional embedding represented as $\varepsilon$ measures the similarity between $x_1$ and $x_2$ pairs of points expressed as:

$$Q_{ij} = \frac{\left(1 + \|x_i - x_j\|^2\right)^{-1}}{\sum_{k \neq 1}\left(1 + \|x_k - x_l\|^2\right)^{-1}} \tag{3}$$

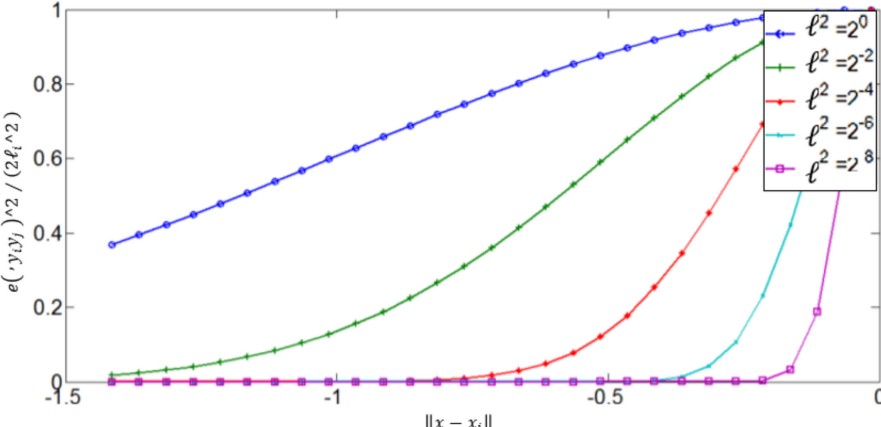

**Figure 3.** Gaussian Kernel optimization values. The optimal value of the Gaussian kernel differs at every point, and its value decreases in areas that the density of the data is higher in the data space and it increases in regions of lower data density, respectively.

From Equation (3) above, the student-t distribution enables the identification of dissimilar variable input $y_i$ and $y_j$ at the low-dimensional counterparts $x_i$ and $x_j$. Applying

the Kullback–Leibler principle [73], we compute the divergence existing between the *Q* and P express as follows:

$$K_u\ell = K_l(P\|Q) = \sum_{i\neq j} P_{ij} \log \frac{P_{ij}}{Q_{ij}} \tag{4}$$

The $K_l$ in Equation (4) above is the Kullback–Leibler normalization factor [73] between *P* and *Q* distribution over the number of distinctive variables. To estimate the input variables similarity computed, the approximation technique of Barnes–Hut cited in [74] was applied. This technique also helps to avoid embedding quality from being negatively impacted while locating the nearest neighbors for every *N* number of input variable and enables seen the relationship between two dimensions with size and color (see Figure 4) similar to bubble chat [75]. By applying the approximation technique, Equation (1) was reformulated as:

$$P_{ij} = \begin{cases} \dfrac{E\left(-e\left(y_i,y_j\right)^2}{2\ell_i^2\right)}}{\dfrac{\sum_{k\in N_i} E\left(-e\left(y_i,y_k\right)^2}{2\ell_i\right)}}, \; \textit{Provided j belongs to } N_i \end{cases} \tag{5}$$

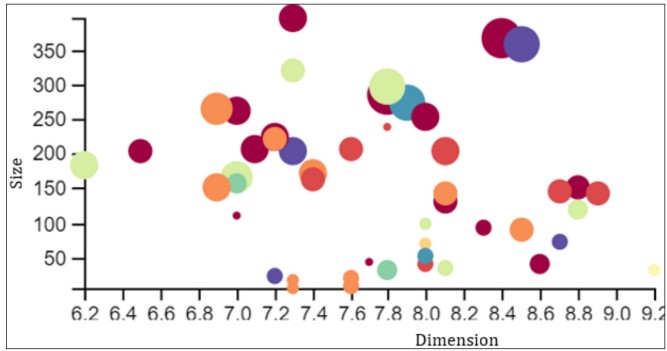

**Figure 4.** Points correlation and proportions.

Equation (5) forms the vantage point tree VP-tree [76,77] for retrieving permission features from the data set without necessarily being in a high dimensional variable space based on the $e(y_i, y_j) = \|y_i, y_j\|$ metric function. The VP-tree technique enables our method to be applied on data sets whose $x_i \in \mathbb{R}^D$ vectors have input data of high dimensions.

*4.3. The Self-Organizing Map (SOM)*

As highlighted in Section 4, SOM is an important tool for creating and representing highly dimensional input space, which binds correlated variable densities and displays them using colors or the variable region. Another advantage of this algorithm is that it can encapsulate complex data and still maintain the topographical attributes of the data in the input vector space [78]. Consider a neural network of *b* weights consisting of *y* input correlating to an Android app having its permissions represented by binary string $\{0,1\}$. Let *g* represent the matching neuron; then, the Euclidian distance existing between *y* and *b* is expressed as:

$$e_d = \|y - b_g\| = min_i\{\|y - b_i\| \tag{6}$$

$$b_i(h+1) = b_i(h) + r_{gi}(h)[y(h) - b_i(h)] \tag{7}$$

Equation (7) was achieved by adjusting matching neuron *g* in the neighbor neuron index *i* and the kernel $r_{gi}$ and time *h*. The function $r_{gi}$ is a function with reference to the

Euclidian distance and time. If the learning rate of the model is $\alpha$ and the location of the two-dimensional network neuron is $(h)$, Equation (7) becomes:

$$r_{gi}(h) = r\left(\|l_g - l_i\|, h\right)\alpha(h)r_{gi}(h) = E\left(-\frac{\left(\|l_g - l_i\|^2\right)}{(2\ell^2(h))}\right)\alpha(h) \tag{8}$$

where $\ell$ is the Gaussian kernel factor (Section 4.1, Equation (1)). During the network training, the learning rate of 0.01 with a batch size of 8 was set on the weight decay of 0.001 at momentum of 0.9. We set the value of $\alpha(h)$ high to enhance better transformations in the Self-Organizing Map weight points (Figure 5) and the U-matrix (Figure 6) of the permissions. However, the limitation of setting a high value of the learning rate is that changes in the network neurons were not incremented significantly. Consequently, the training parameters must be tested using exploratory analysis before permission can be visibly visualized.

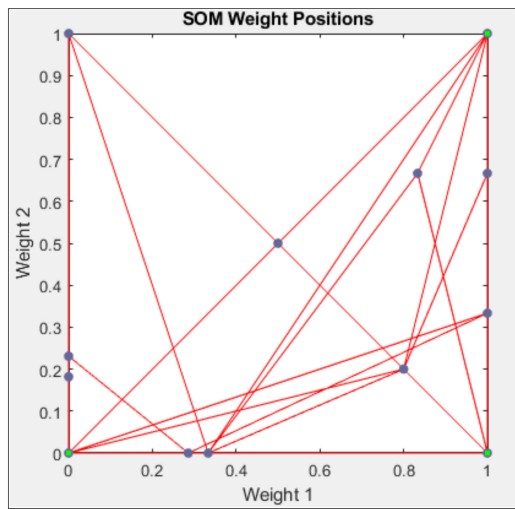

**Figure 5.** SOM weight positions of the permission input variables.

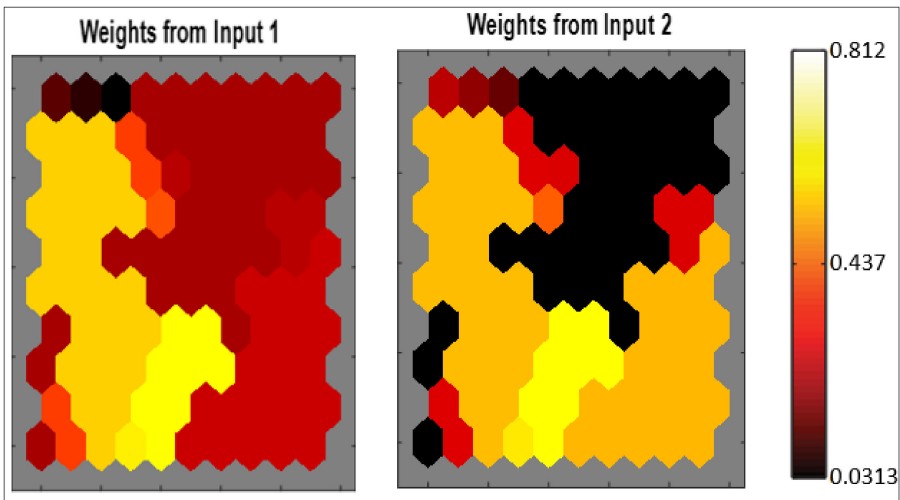

**Figure 6.** SOM U-matrix for Android permissions. The hexagons indicate spaces between network neurons with a binary string $\{0, 1\}$, which forms the Euclidean distance.

As shown in Figure 6, there are some regions (in the hexagon upper region) that are darker than others, while others are sparsely populated involving. During the inspection,

we discovered that permissions presented in the darker regions are those that use traditional mobile features such as text messaging and calls, among others (see Section 4.4). This shows that those regions have related input patterns implying that Android apps in the same category with permissions in the same region do not have necessarily behave the same and request the same permissions. It then means that Android apps in the same cluster can request similar permissions. It is then significant to understand applications and permission clusters by performing exploratory plane analysis to determine Android permission correlation and how they are frequently requested.

### 4.4. Exploratory Factor Analysis

Determining the correlation between permissions in complex data sets such as Android malware is challenging. A total of 12,267 malicious and 10,837 benign applications were used for our study. Exploratory factor analysis is a great simulation technique in reducing multivariate data dimensionality to identify minute factors that explain the underlying correlation between variables. This technique was applied in the research of [79] to analyze the dataset. We apply the equamax exploratory factor analysis rotation method to reduce the number of independent *k* factors based on their eigenvalues (see Figure 7) that could be obtained from many correlated permissions and to test for possible permission correlation using correlation methods (Pearson, Spearman, and Kendall correlation).

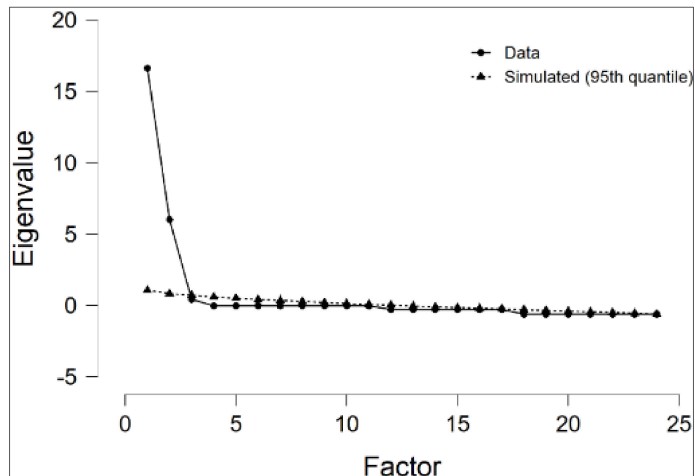

**Figure 7.** The Scree plot shows how much there is variance in the dataset by eigenvalues in the three factors.

Factor analysis technique enhances establishment of the best portion of permissions variables interrelationship among factors. Using Kaiser–Meyer–Olkin's approach [80], we extracted three factors by MSA extraction procedure. The factors in the structure matrix of the selected variables using the equamax rotation method (see Table 2) explain the correlation among the permission variables. Each of the factors (factor 1, factor 2, and factor 3) was represented by circles, and the permission variables selected were represented by boxes in the path diagram (see Figure 8).

**Table 2.** Structure Matrix of some selected variables using equamax rotation method.

| | Factor 1 | Factor 2 | Factor 3 |
|---|---|---|---|
| WRITE_SMS | 0.972 | 0.077 | −0.220 |
| READ_SMS | −0.252 | 0.967 | |
| SEND_SMS | 0.934 | −0.302 | 0.191 |
| RECEIVE_SMS | 0.972 | 0.077 | −0.220 |
| RECEIVE_WAP_PUSH | −0.252 | 0.967 | |
| CALL_PHONE | 0.934 | −0.302 | 0.191 |
| READ_PHONE_STATE | −0.252 | 0.967 | |
| READ_CALL_LOG | 0.934 | −0.302 | 0.191 |
| WRITE_CALL_LOG | 0.972 | 0.077 | −0.220 |
| ADD_VOICE_MAIL | −0.252 | 0.967 | |
| USE_SIP | 0.934 | −0.302 | 0.191 |
| PROCESS_OUTGOING_CALLS | 0.972 | 0.077 | −0.220 |
| GET_ACCOUNTS | −0.252 | 0.967 | |
| READ_ACCOUNTS | 0.934 | −0.302 | 0.191 |
| WRITE_ACCOUNTS | 0.972 | 0.077 | −0.220 |
| ACCESS_CAMERA | −0.252 | 0.967 | |
| READ_EXTERNAL_STORAGE | 0.934 | −0.302 | 0.191 |
| WRITE_EXTERNAL_STORAGE | 0.934 | −0.302 | 0.191 |
| READ_CALENDAR | 0.934 | −0.302 | 0.191 |
| ACCESS_COARSE_LOCATION | −0.252 | 0.967 | |
| ACCESS_FINE_LOCATION | 0.972 | 0.077 | −0.220 |
| RECORD_AUDIO | 0.934 | −0.302 | 0.191 |
| WRITE_CALENDAR | 0.972 | 0.077 | −0.220 |

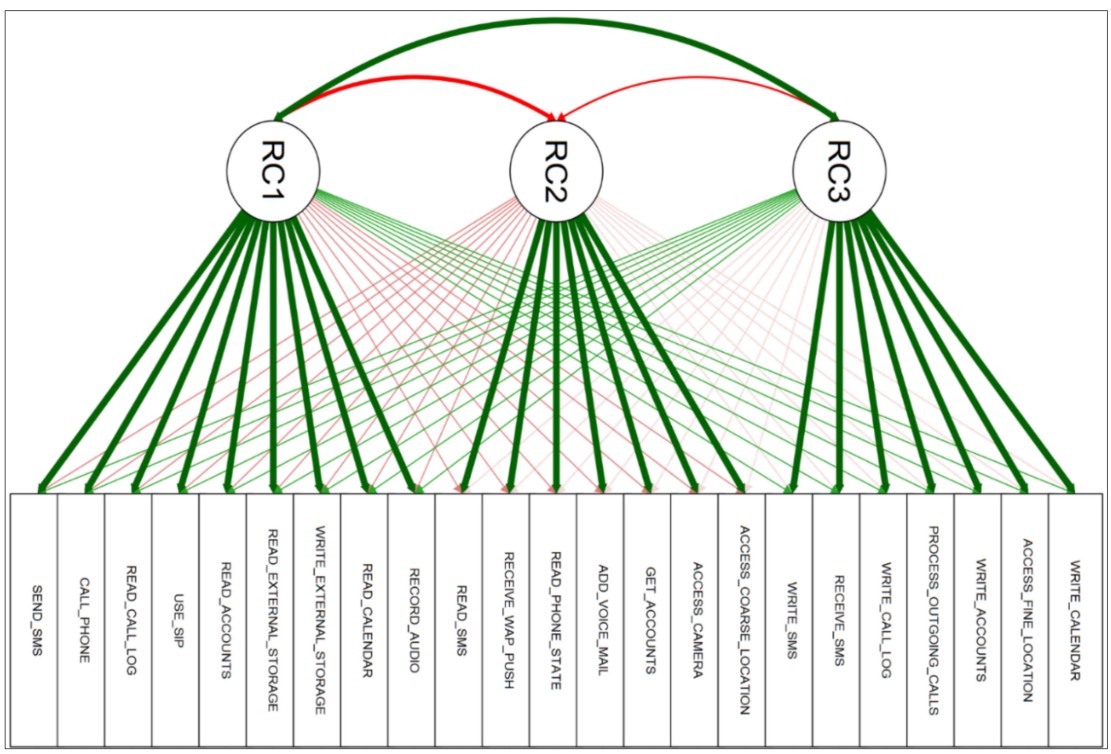

**Figure 8.** Exploratory analysis path diagram of the correlation factors and the permission variables.

Figure 8 shows the cursors from the factors pointing to the permission variables signify how a factor loads on the permission variable. When there is a positive loading, the model represents it with green arrows; otherwise, red arrows. The dimension of the loading is determined by the size of the arrow. This means that when loading is high, the arrows become wider; when loading is low, the arrows become narrower. Ac-

cordingly, record_audio, read_calendar, write_external_storage, read_external_storage, read_accounts, use_sip, read_call_log, call_phone, and send_sms are associated with factor 1. Read_sms, receive_wap_push, read_phone_state, add_voice_mail, get_accounts, access_camera, and access_coarse_location are associated with factor 2. Lastly, write_calender, access_fine_loaction, write_account, process_outgoing_calls, write_call_logs, receive_sms, and write_sms are associated with factor 3. We observed that though there is a clear association of specific variables to specific factors, some of the variables are partly associated with more than two factors.

Figure 9 shows the exploratory factor analysis visualizing some traditional permissions frequently requested by Android applications. The result shows that most permissions in the large subcategory were requested by very few Android applications, while frequently used permissions were those in the small subcategory. With regards to state of the art, this signifies that there is no adequate expressiveness from permissions that are requested frequently. We infer that those not frequently requested could be disintegrated into the common class. As a result, we suggest adding finer granularity as a security approach for Android permissions that are frequently requested by applications will enhance this expressiveness and enhance Android security, especially when combined with the ones occasionally requested.

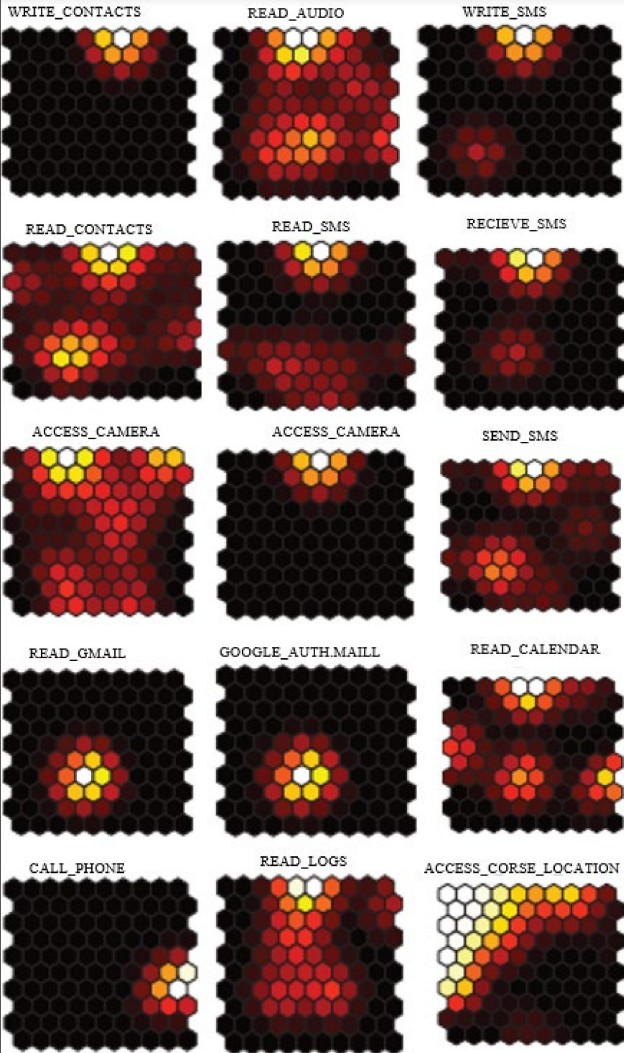

**Figure 9.** Exploratory visualization of permission.

*4.5. Correlation between Android Permissions*

According to [81], when modeling relationships between variables of interest, Bayesian inference is the best starting point to formulate the prior distribution of such a model. Bayesian prior distribution enables quantization of the parameter values before sample size observation. It reflects the observed attributes of the *p*-values and gives more information about the data set central tendency that forms parameter estimation points. Posterior differences quantify the parameter estimation uncertainties, which indicates that the uncertainty increases with an increase in the dispersion. However, it is not every correlation coefficient that could determine the posterior distribution from sample sizes as identified by [82].

In this subsection, we aim to represent the interaction between permission variables as a factor of linear combinations of threat and permission levels. Let $X$ represent a permission variable relating to whether the threat level in the permission variable is high ($X = 1$) or low ($X = 0$). Additionally, let $\varnothing$ be the parameter symbolizing the probability that $X = 1$, that is, the threat level of the dangerous permission is high. Going by the Bayesian correlation approach, we compute and express our confidence in the probability $\alpha$ using the posterior distribution on parameter $\varnothing$. Following the Bayesian correlation approach, we assume that the parameter $\varnothing$ obeys the generalized hyperparameters Beta distribution [83] of the prior expectation and variance control of the $\varnothing$ prior. Let $\beta$ represent Beta consisting of the prior expectation $e$ and variance control $g$ of the prior given as:

$$\beta(eg,\ e(g-1)) \tag{9}$$

Using empirical evidence [84,85], consider $\bar{\epsilon}$ to be the empirical evidence of dangerous permission in a malicious file. Let $\bar{\epsilon} = 1$ if the threat level of any dangerous permission is high in the data set, and $\bar{\epsilon} = 0$ contrarily. If no benign permission variable is present in the list of permission features, then no threat level will be observed. Otherwise, it will be detected by the probability $\alpha$. The evidence for the presence of a high threat level in the model is expressed as $H_1$, otherwise $H_0$. The evidence is captured by delta $\beta$, which depends on the decision. Let $\alpha \in \mathbb{R}^+$ and $\beta \in \mathbb{R}^+$ be the parameters for the probability density function, which supports $x \in [0,1]$. The moments of the empirical evidence $\bar{\epsilon}$ and variance $\sigma^2$ for each permission variable obtained from the dataset expressed as:

$$\bar{\epsilon}(X) = \alpha(\alpha + \beta)^- \tag{10}$$

$$\sigma^2(X) = \alpha\beta(\alpha + \beta)^{-2}\quad (\alpha + \beta + 1)^{-1} \tag{11}$$

To assess the goodness of fit of the cumulative distribution function of a permission variable, we applied Cramer–von Mises statistical parameterization approach [86]. In estimating the parameters, the range of $x$ was set from 0 to 1 $x \in [0,1]$ with a confidence interval of 95%. The range is the highlighted interval for both the density and the probability (as shown in Figure 10).

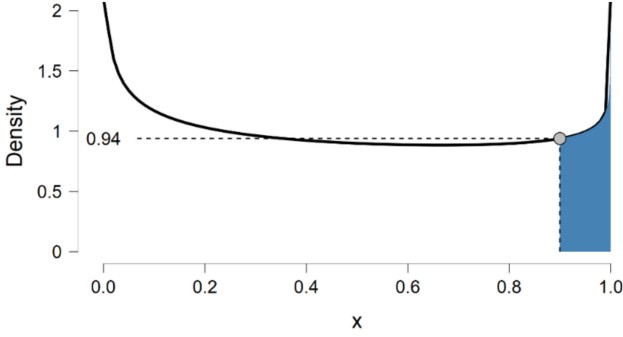

**Figure 10.** Density plot for random permission variable. The y-axis displays the value of the density function for a particular value of the permission variable. The dotted line represents the density, while the blue-colored region is the probabilistic region.

In the analysis, we chose to reflect on the risk permission request poses on mobile devices, particularly the Android platform. We focus on analyzing the relationship between the threat and protection level. However, we are uncertain of the probability that a permission request has a high threat level or low protection level. The decision to set $\propto$ over the parameters was based on the robust Bayesian approach [87] of analyzing uncertainties in variables correlation, assuming that the variance is homogenous in the variable distribution. Given that the variance treatment is equal:

$$\sigma^2{}_1 = \sigma^2{}_2 = \cdots = \sigma^2{}_k \tag{12}$$

$$\sigma^2{}_i \neq \sigma^2{}_j \tag{13}$$

where k is the number of permission groups in the sample distribution and i and j represent at least one pair. To check for sample deviation from the normal distribution, we applied Bartlett's test assumption check [88]. Let $\sigma_P^2$ represent the pooled variance expressed as:

$$\sigma_P^2 = \frac{\sum_{i=1}^k (N_i - 1)\sigma_i^2}{(N - k)} \tag{14}$$

$$\chi^2 = \frac{(N - k)|_n\left(\sigma_P^2\right) - \sum_{i=1}^k (n_i - 1)|_n\left(\sigma_{i)}^2\right)}{1 + \frac{1}{3(k-1)}\left(\sum_{i=1}^k \left(\frac{1}{n_i - 1}\right) - \left(\frac{1}{N-k}\right)\right)} \tag{15}$$

Equation (7) represents Bartlett's test. $\sigma_P^2$ represent the pooled variance, $\sigma_i^2$ is the variance of the $i^{\text{th}}$ permissions groups, N is the permission sample size, $n_i$ is the permission sample size in the $i^{\text{th}}$ variable group, and $k$ is the number of factors with $k$-levels. The result of Bartlett's test shows the interaction of the effects of permission sample size and other parameters (see Table 3).

**Table 3.** Interaction effects of the test statistics using MANOVA.

| *n* | $\chi^2$ | *k* | $\sigma^2$ |
|-----|----------|-----|-----------|
| *n*1 | 15.925 | 2.130 | 0.708 |
| *n*2 | 16.002 | 3.403 | 0.611 |
| *n*3 | 16.201 | 3.342 | 0.579 |
| *n*4 | 15.925 | 2.133 | 0.731 |
| *n*5 | 16.002 | 2.421 | 0.816 |
| *n*6 | 15.001 | 3.342 | 0.903 |
| *n*7 | 15.521 | 2.530 | 0.881 |
| *n*8 | 15.106 | 3.501 | 0.908 |
| *n*9 | 17.067 | 2.367 | 0.736 |

If the correlation that exists between Android permissions is represented as $\rho$, consider permission variables with respective prior odds represented as $p(u_0)|p(u_1)$ in the dataset represented as $D = \{y_1, y_2, \ldots, y_N\}$. The model posterior odds are expressed as:

$$\frac{p(u_0|D)}{p(u_1|D} = \frac{p(D|u_0)}{p(D|u_1)} \times \frac{p(u_0)}{p(u_1)} \tag{16}$$

where $u_0$ and $u_1$ represents instantiations for the null hypothesis ($H_0$) and alternative hypothesis ($H_1$), respectively. From Equation (18), the function $\frac{p(D|u_0)}{p(D|u_1)}$ forms the Bayes factor. This is the proportion of the likelihoods of the posterior distribution. However, we represented the function (Bayes factor) in our model as $Bf_{\mathbf{n}}$ where $\mathbf{n}$ takes values of $_{10}$ and $_{01}$ of the permission sample size in the $i^{\text{th}}$ variable group expressed as:

$$Bf_n = \frac{p(u_0|D)}{p(u_1|D} \times \frac{p(u_1)}{p(u_0)} \tag{17}$$

These factors represent the two sides of the research hypothesis at a stretched beta prior value range of $-1$ to 1 using density ratio. Our Bayesian correlation model was formulated using Pearson's correlation coefficient represented as $\rho$. At $\rho = 0$ under $Bf_{01}$ and posterior under $H_1$ expressed as:

$$\text{Bf}_{01} = \frac{p(D|H_0)}{p(D|H_1)} = \frac{p(\rho = 0|D, H_1)}{p(\rho = 0|H_1)} \tag{18}$$

$$\text{Bf}_{10} = \frac{p(D|H_1)}{p(D|H_0)} = \frac{p(\rho = 1|D, H_0)}{p(\rho = 1|H_0)} \tag{19}$$

In the alternative hypothesis, $u_1$ instantiates the presence of correlation between permission requests variables represented as $H_1: \rho \neq 0$. Using Jeffrey's prior [89], we assign prior distribution correlation under the null hypothesis to take uniform default values between the range $-1$ to $+1$ as a default $\rho$. This was relative to the alternative hypothesis using two population correlation coefficients, namely Pearson's rho and Kendall's tau. The two correlation coefficients were used for the posterior and prior comparison of our model (as shown in Figure 11).

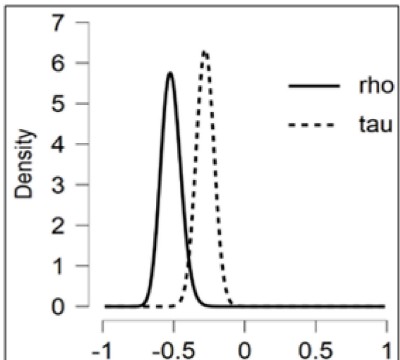 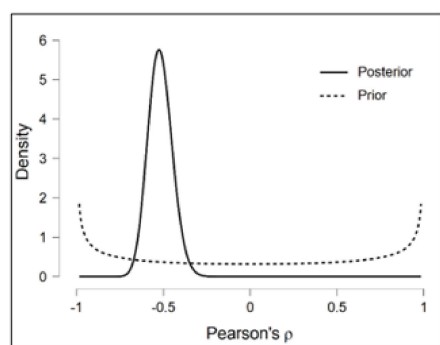

**Figure 11.** Sequential analysis of the empirical evidence of dangerous permission with high threat levels.

4.5.1. Pearson Correlation

The non-parametric function in our study helps to measure how permission request variables correlate to each other. Given x and y as permission variables contained in the Android malware dataset, the Pearson correlation is expressed as:

$$r = \frac{n\Sigma x_i + y_i - \Sigma x_i + \Sigma y_i}{\sqrt{n\Sigma x_i^2 - (\Sigma x_i)^2} \sqrt{n\Sigma y_i^2 - (\Sigma y_i)^2}} \tag{20}$$

The $n$ represents the sample size in the dataset distribution. In determining $r$, we assumed that all the permission request variables in the dataset were normally and equally distributed according to the principle of variable homoscedasticity [90–92]. Otherwise, the principle of variable heterogeneity [93] sets in. Let $\tau$ represent the heterogeneity that occurs in the Android data set $D$. We represent the group-level effect size with $\mu$. The Bayes factor $Bf_n$ then propagates a heterogeneity factor written as $Bf_{rf}$ and $Bf_{fr}$ under prior $H_1$ and random factor $H_1$ (as shown in Figure 12). At this instance, the $Bf_{10}$ gives the Bayesian inference while $Bf_{01}$ is automatically set to zero. This helps in accepting or rejecting the null hypothesis. We assumed that when heterogeneity occurs in the permission distribution, the Bayes factor becomes heterogeneous for $H_1$. Table 4 shows the posterior estimate of the model.

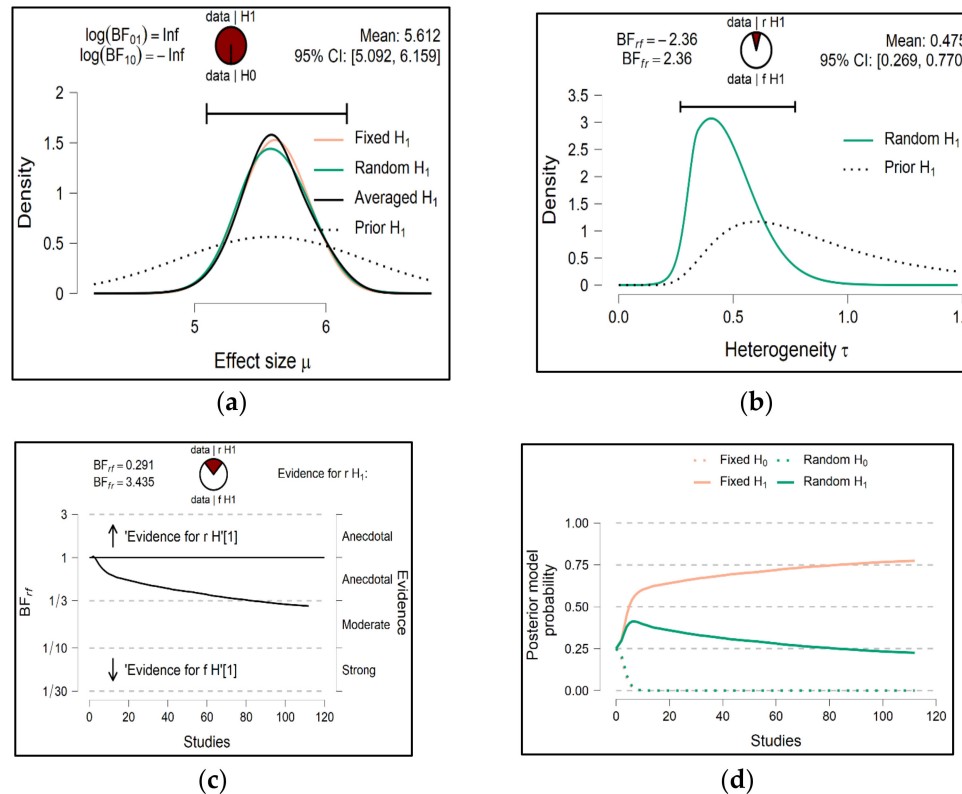

**Figure 12.** Flag supported correlation when permission distribution is heterogeneous in the sample distribution. (**a**) Model Effect size. (**b**) Heterogeneity. (**c**) Bayes factors heterogeneity. (**d**) Posterior model probabilities.

**Table 4.** Posterior estimate of the model.

|  |  |  | Mean | SD | Lower | Upper | $BF_{10}$ |  |
|---|---|---|---|---|---|---|---|---|
| Fixed effects | μ |  | 5.585 | 0.274 | 5.030 | 6.117 | $2.8 \times 10^{84}$ |  |
| Random effects | μ |  | 5.588 | 0.276 | 5.049 | 6.124 | $7.2 \times 10^{51}$ |  |
|  | τ |  | 0.209 | 0.163 | 0.010 | 0.601 | 0.291 | [a] |
| Averaged | μ | [b] | 5.586 | 0.273 | 5.045 | 6.119 | ∞ |  |
|  | τ | [c] |  |  |  |  | 0.291 |  |

Where μ is the group-level effect size which forms the mean in the distribution. [a] Bayes factor of the random effects $H_1$ over the fixed effects $H_1$. Posterior estimates are based on the models that assume an effect to be present. The Bayes factor is based on all four models: fixed effects $H_1$ and random effects $H_1$ over the fixed effects $H_0$ and random effects $H_0$. [c] Model averaged posterior estimates for τ.

From the data set variables, selecting four of the Android permissions requests: READ_SMS, CALL_PHONE, RECEIVE_SMS, and WRITE_SMS of n total sample population of the permission variables, the Pearson *r* was determined using the *p*-value and the VS-MPR (https://doi.org/10.17862/cranfield.rd.13363322.v1 (accessed on 2 August 2021)). As shown demonstrated in Figure 13, Pearson *r* takes a value range between −1 and 1. The *r* value measures the potency of the correlation between the data variables under examination. Using the rule of Thumb postulated by Guildford [94] to determine the degree of correlation between permission requests, our research states that if the value of *r* is greater than 0, there is a positive correlation between the Android permission variables.

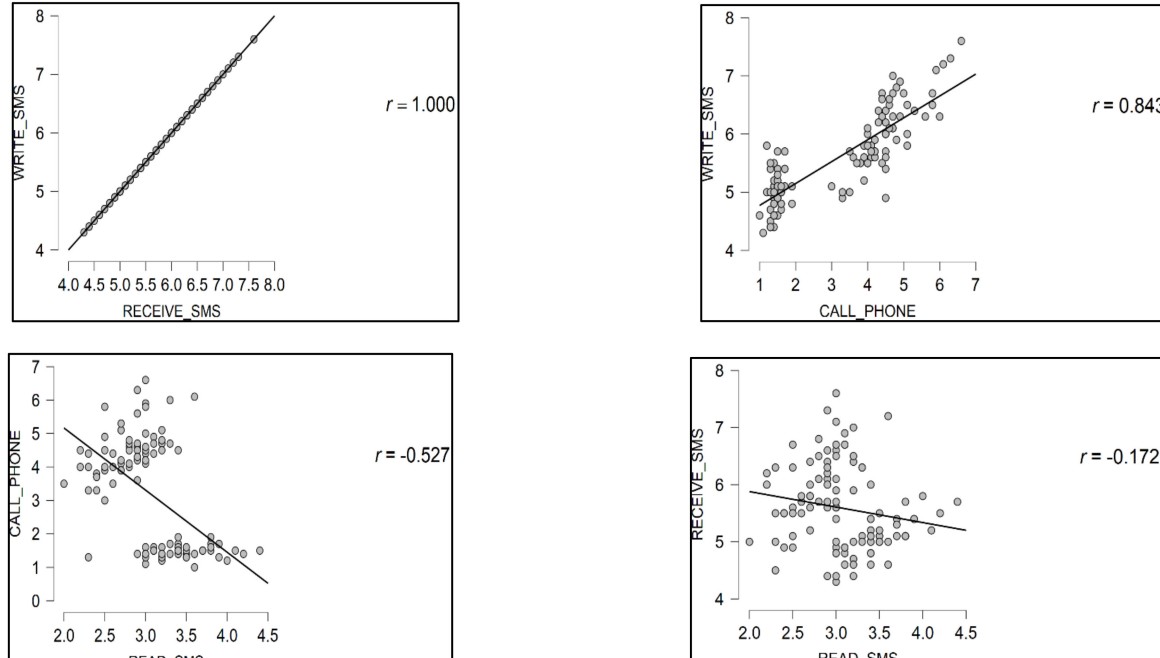

**Figure 13.** Pearson's correlation coefficient for some selected permission variables.

If the value of *r* is less than 0, then there is a negative association that exists between the threat level and the protection level. However, if *r* turns close to 0, it implies that the correlation existing between permissions is weak. The correlation between two permissions is considered very strong if the value of *r* is greater than 0.7. In a situation where the correlation between the permissions request is negatively correlated, the result of the *p*-value is less than 0.001, expressed as *p* < 0.1.

### 4.5.2. Spearman Correlation

A non-parametric check measures the level of magnitude of how two or more permissions requests in the dataset are associated with one another. However, our research only focused on determining the degree of correlation between two permissions and not multi-permissions variables association. We assumed that the scores in each permission request are uniformly associated with other permission requests. Given the sample size *n* and the rank difference $d_i$ between the corresponding permission variables, the Spearman correlation p, between two permission requests is expressed as:

$$\rho = \frac{1 - 6\Sigma d_i}{n(n^2 - 1)} \tag{21}$$

Let us consider some permission requests from the sample space *n* = 112. The degree of ranking between any of the two permission variables helps in ordering the observation of their correlation magnitude. The ranking order demonstrates that the higher the *p*-value existing between two permission requests, the lower the value of Spearman ρ and vice versa. If Spearman ρ and the *p*-value are inversely proportional, the correlation between the threat level and the protection level of a permission request is inversely proportional to each other (https://doi.org/10.17862/cranfield.rd.13363322.v1 (accessed on 2 August 2021)). Figure 14 shows Spearman's rho correlation coefficients of some selected variables.

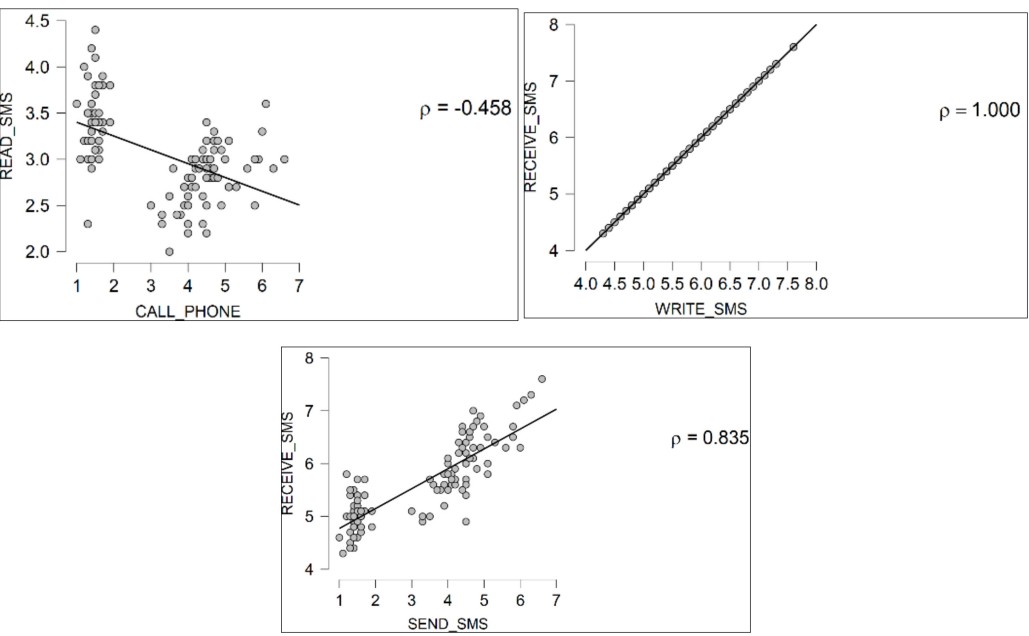

**Figure 14.** Spearman's rho correlation coefficients.

### 4.5.3. Kendall Correlation

Kendall correlation measures the stability of how two permission variables depend on each other. For instance, if we consider four permission variables say: WRITE_SM, SEND_SMS, RECEIVE_SMS, and READ_SMS with a sample size *n*, according to [95,96], the composite number of matches between two variables can be expressed as $n(n-\mathrm{i})/2$. The Kendall correlation between the variables can be expressed as:

$$\tau = \frac{n_c - n_d}{1/2(n-1)} \tag{22}$$

where $n_c$ *and* $n_d$ represent the number of concordance and discordance, respectively. The number of concordances ensures an even ordering of permission variables while discordance orders permission variables heterogeneously or abnormally. The *p*-value took a range of $p < 0.05\ p < 0.01\ p < 0.001$. The -Sellke Maximum *p* –Ratio (VS-MPR) was Based on the *p*-value, where the maximum possible odds in favor of H$_1$ over H$_0$ equals 1/(-e *p*log(*p*)) for $p \leq 0.37$, as shown in Figure 15.

As shown in Figure 15, it is only the correlation between RECEIVE_SMS and WRITE_SMS that has $\tau$ = 1. It, therefore, means that it is only the correlation between RECEIVE_SMS and WRITE_SMS that captures the exact non-linearity that exists in all the four selected permission request variables. The stability of dependence tends to increase as the sample size increases. When correlating a few variables, there is no significant difference in the appearance of the scatter plots of the plotted variables. However, the difference becomes clearer when the sample size increases (https://doi.org/10.17862/cranfield.rd.13363322.v1 (accessed on 2 August 2021)).

### 4.6. Comparison of the Correlation Coefficients

This sub-section of the research paper compares the correlation coefficients under study: Pearson, Spearman, and Kendall's tau-b. The comparison aims to test the statistical significance to assess if the critical value is less than the observed value. This will help in rejecting the null hypothesis under study. During this study, we observe that the correlation line is difficult to identify at extreme negative values example, $r = -0.172$, $\rho = -0.243$, and $\tau = -0.120$. This shows that a correlation is weak in those instances. However, it becomes more visible and clearer as the correlation becomes stronger. The order of permissions requests in the correlation is not significant but provides the only evidence of association,

not causation. As shown in Figure 16, the positive values and the negative values indicate a positive and negative association between the permission requests, respectively.

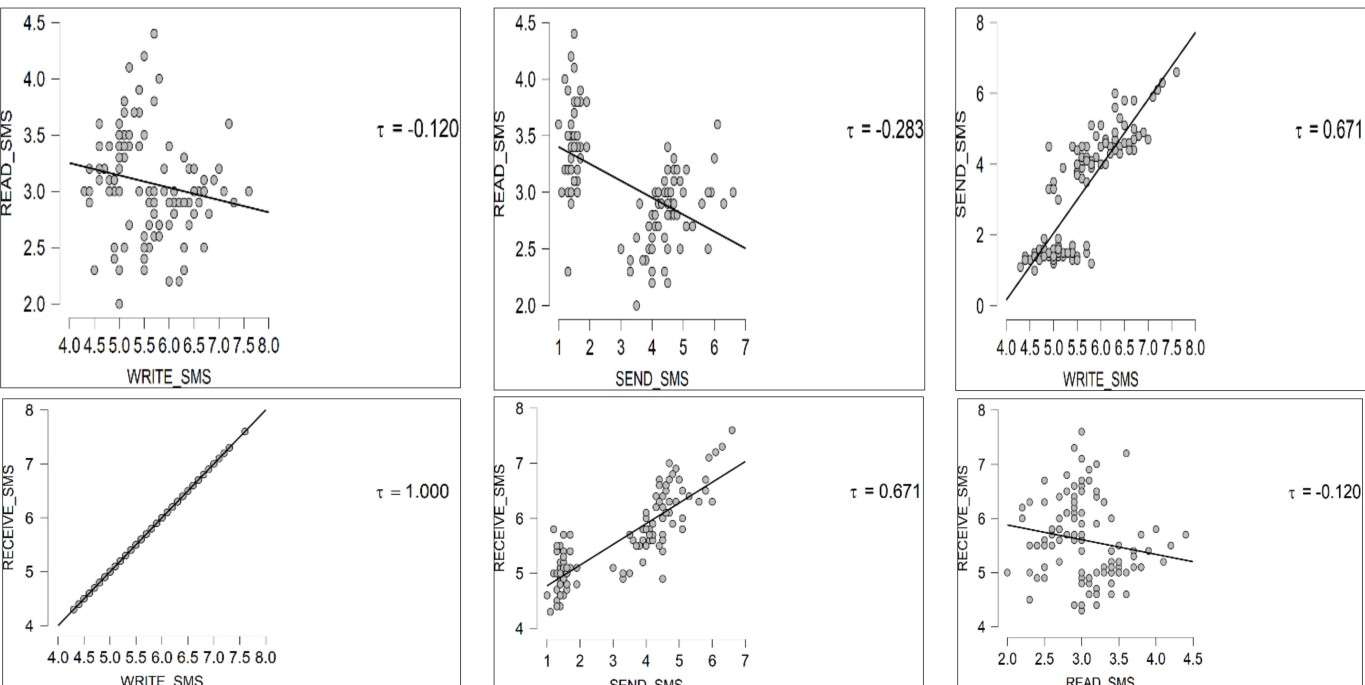

**Figure 15.** Kendall's tau-b correlation coefficient for some selected permission variables.

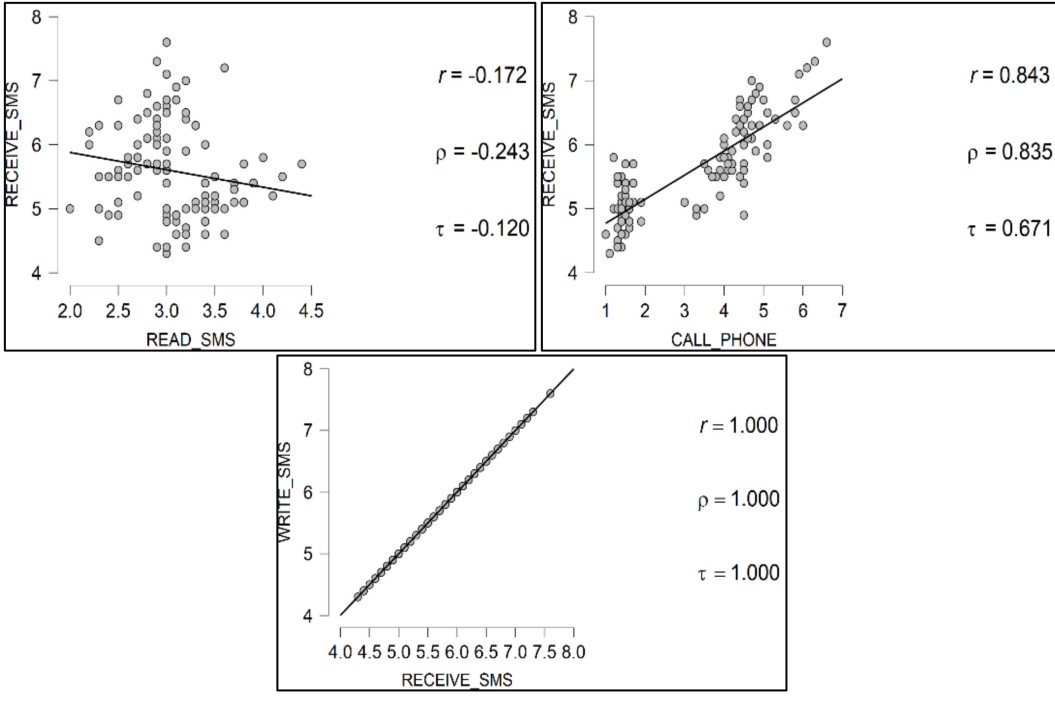

**Figure 16.** Correlation at different degrees of coefficient strengths.

Extreme observations are affected and strongly impacted by the correlation coefficient. This signifies that the value of extreme values decreases as the number of samples increases. This information is necessary to help us conduct further analysis to provide additional information about the relationship between threat level and protection level of each permis-

sion request. The overall correlation table that compared the results of the three correlation coefficients generated by our model (https://doi.org/10.17862/cranfield.rd.13363322.v1 (accessed on 2 August 2021)). The correlation heatmap for Spearman's rho, Pearson's r correlation, and Kendall's tau-b correlation heatmaps (https://doi.org/10.17862/cranfield.rd.13363322.v1 (accessed on 2 August 2021)).

*4.7. Threat and Protection Levels Evaluation*

A threat model can be defined as a well-defined description regarding the information which affects or influence an application's security. Modeling a threat level involves information capturing, organization, and analysis to facilitate decision-making concerning the security threat of an application. Modeling threat helps to identify vulnerabilities and other risks associated with an application or a system. We acknowledge that dangerous permissions as threats are hard to address apart from adjusting to their requirements. Privacy threats caused by dangerous permissions create a serious security risk to users and system developers.

This subsection of the research focuses not on developing a threat model but on determining the relationship between threat and protection level in some selected Android permission requests. Understanding the threat level of permission is significant to classify permission to be normal or dangerous. The implication is that if the permission access granted has not much negative effect on that resource; such a permission request has a low threat level while the device has high protection level at that instance. Consider a malicious Android application requesting permissions to access the camera resource of the mobile device when the user does not open a feature that necessitates using the camera, it looks so suspicious. Using Bayesian hierarchical modeling was deployed in our research to estimate the relationship between protection and threat level in each permission request by an Android malicious application. In our approach, we defined the two concepts as Level 1 for protection level and Level 2 for threat level, respectively.

Level 1: Protection level. Protection level indicates the possible risk suggested in Android permission and the process a device should adhere to when deciding whether allow an application access system resources. In our model, it is represented as the observed data in the distribution of the permissions selected from the sample space. Let $\varphi$ and $\partial$ denote true variance values, and $\varphi_\varepsilon$ and $\partial_\varepsilon$ represent their respective errors associated with their distribution. We then state that for each $i^{th}$ observation taken values from 1 to $N$ represented as $i = 1 \ldots N$, their observed data values assume values with respect to their errors and variances respectively written as:

$$\hat{\varphi}_i \sim \text{Normal}\left(\varphi_i, \sigma^2{}_{\varphi_{\varepsilon i}}\right) \tag{23}$$

$$\hat{\partial} \sim \text{Normal}\left(\partial_i, \sigma^2{}_{\partial_{\varepsilon i}}\right) \tag{24}$$

From Equations (15) and (16), $\sigma^2{}_{\varphi_{\varepsilon i}}$ and $\sigma^2{}_{\partial_{\varepsilon i}}$ represent the assumed know priori error variances for $\varphi$ and $\partial$. The error variances across the observation $N$ do not assume homogeneity, but each of the $i^{th}$ observations assumed their unique individual error variances.

Level 2: Threat level. This is meant to provide a distinct indication of the probability of a malicious attack on a mobile device. In our model, threat level represents the inferred parameters for $\varphi_i$ and $\partial_i$ observations. The informed parameters assumed bivariate normal distribution with their respective means and variances. Let the mean and variance parameters be defined as Mean: $\mu \in \mathbb{R}$; Variance: $\sigma^2 \in \mathbb{R}^+$ with a support $x \in \mathbb{R}$ where R assumes the value $\{0, 1\}$ properly written as $x \in \{0, 1\}$. The two parameters are functions of the two moments $E(X) = \mu$ and $Var(X) = \sigma^2$, respectively. Additionally, let $\mu_\varphi$ and $\mu_\partial$ be the mean for the inferred parameters $\alpha$, which forms the effect size of the distribution given as:

$$\mu = \left(\frac{\mu_\varphi}{\mu_\partial}\right) \tag{25}$$

The covariance confusion matrix estimated from the prior distribution of the selected sample set is given as:

$$Cov(\varphi, \partial) = \begin{pmatrix} \sigma^2_\varphi & \rho\sigma\varphi\sigma_\partial \\ \rho\sigma\varphi\sigma_\partial & \sigma^2_\partial \end{pmatrix} \tag{26}$$

The values of the prior of $\mu_\varphi$ and $\mu_\partial$ are set to be large $(-1, 1)$ according to Jeffreys' principles [97] to generate no uniformity for its distribution while the prior for the variance, $\sigma_\varphi$ and $\sigma_\partial$ are equally set with the same dimension to prior distribution uninformative. The advantage of setting in this dimension is to enhance the automatic adjustment of $\rho$ when a new data variability source that could result in observations uncertainty is added to it. Specifically, $\hat{\varphi}$ and $\hat{\partial}$ observations shrink in the direction of their matching class mean. In the model, if the observed parameters are highly dispersed than the inferred parameters, then permission should consider having a higher threat level than the protection level.

The relationship between level 1 (protection level) and level 2 (threat level) with data density distribution using RM factor. The result of the correlation between the observed (protection level) and the inferred (threat level) as shown in Figure 17. This demonstrates that the posterior distribution of the observed correlation is decreased to lower values compared to the inferred correlation of the model correspondingly. Figure 17 shows the correlation between the represented levels estimated at $\varphi_\varepsilon$ and $\partial_\varepsilon$, respectively. The posterior representation and the model accuracy in the estimation of the inferred and the observed parameters. The summary of the evaluation metrics for the model's precision, recall, F1 score, support, and the AUC is represented in Table 5, while the results of the model class proportion for the observed and inferred parameters are represented in Table 4 accordingly.

Figure 18 is the matrix plot for correlation between protection and threat level existing between permissions. The red color indicates the threat level while the blue indicates the protection level of individual permissions, which is distributed sparsely in an application. The denser region shows that the relative correlation between the threat and protection level between the two permissions is strong. One of the remarkable findings is that each permission has a protection and threat level. We also identified that Android permissions with the same protection level have the same threat level. This signifies that such permissions have a similar component plane in their distribution in the sample space. The details of the correlation model comparison of some selected permissions are presented in Table 6.

**Table 5.** Evaluation metrics.

|  | Precision | Recall | F1 Score | Support | AUC |
|---|---|---|---|---|---|
| Protection level | 0.917 | 1.000 | 0.957 | 11 | 0.996 |
| Threat level | 1.000 | 0.909 | 0.952 | 11 | 0.975 |

**Table 6.** Correlation model comparison of some selected permissions.

| Models | P(M) | P(M ǀ data) | BF $_M$ | BF $_{10}$ | error % |
|---|---|---|---|---|---|
| RM Factor 1 + Correlation + SEND_SMS + RECEIVE_SMS + RM Factor 1 ✳ Correlation | 0.050 | 0.966 | 535.883 | 1.000 |  |
| RM Factor 1 + Correlation + RECEIVE_SMS + RM Factor 2 ✳ Correlation | 0.050 | 0.034 | 0.674 | 0.035 | 0.070 |
| RM Factor 1 + Correlation + SEND_SMS + RM Factor 3 ✳ Correlation | 0.050 | $1.044 \times 10^{-21}$ | $1.984 \times 10^{-20}$ | $1.081 \times 10^{-21}$ | 0.537 |
| RM Factor 2 + Correlation + RM Factor 1 ✳ Correlation | 0.050 | $1.005 \times 10^{-29}$ | $1.910 \times 10^{-28}$ | $1.041 \times 10^{-29}$ | 0.979 |
| RM Factor 2 + SEND_SMS + RECEIVE_SMS + RM Factor 2 ✳ Correlation | 0.050 | $3.255 \times 10^{-38}$ | $6.184 \times 10^{-37}$ | $3.370 \times 10^{-38}$ | 0.411 |
| RM Factor 2 + Correlation + SEND_SMS + RM Factor 3 ✳ Correlation | 0.050 | $1.184 \times 10^{-38}$ | $2.250 \times 10^{-13}$ | $1.226 \times 10^{-38}$ | 0.321 |

**Table 6.** *Cont.*

| Models | P(M) | P(M \| data) | BF $_M$ | BF $_{10}$ | error % |
|---|---|---|---|---|---|
| RM Factor 3 + Correlation + RECEIVE_SMS + RM Factor 1 ✳ Correlation | 0.050 | $5.025 \times 10^{-39}$ | $9.547 \times 10^{-38}$ | $5.203 \times 10^{-39}$ | 0.960 |
| RM Factor 3 + RECEIVE_SMS + RM Factor 2 ✳ Correlation | 0.050 | $1.378 \times 10^{-41}$ | $2.617 \times 10^{-40}$ | $1.426 \times 10^{-41}$ | 0.173 |
| RM Factor 3 + Correlation + SEND_SMS + RM Factor 3 ✳ Correlation | 0.050 | $3.116 \times 10^{-49}$ | $5.919 \times 10^{-48}$ | $3.226 \times 10^{-49}$ | 0.265 |
| Correlation + RECEIVE_SMS | 0.050 | $1.790 \times 10^{-13}$ | $3.402 \times 10^{-32}$ | $1.854 \times 10^{-33}$ | 0.007 |
| **Models** | **P(M)** | **P(M \| data)** | **BF $_M$** | **BF $_{10}$** | **error %** |
| SEND_SMS + RECEIVE_SMS | 0.050 | $1.347 \times 10^{-13}$ | $.559 \times 10^{-32}$ | $1.395 \times 10^{-33}$ | 0.580 |
| Correlation + SEND_SMS | 0.050 | $8.379 \times 10^{-14}$ | $1.592 \times 10^{-32}$ | $8.676 \times 10^{-34}$ | 0.885 |
| Correlation + SEND_SMS + RECEIVE_SMS | 0.050 | $3.802 \times 10^{-14}$ | $7.224 \times 10^{-33}$ | $3.937 \times 10^{-34}$ | 0.793 |
| Null model (incl. subject) | 0.050 | $7.182 \times 10^{-13}$ | $1.365 \times 10^{-37}$ | $7.437 \times 10^{-35}$ | 0.529 |
| SEND_SMS | 0.050 | $5.468 \times 10^{-13}$ | $1.039 \times 10^{-35}$ | $5.662 \times 10^{-35}$ | 0.740 |
| Correlation | 0.050 | $1.428 \times 10^{-13}$ | $2.714 \times 10^{-41}$ | $1.479 \times 10^{-35}$ | 0.660 |

Note. All models include subject.

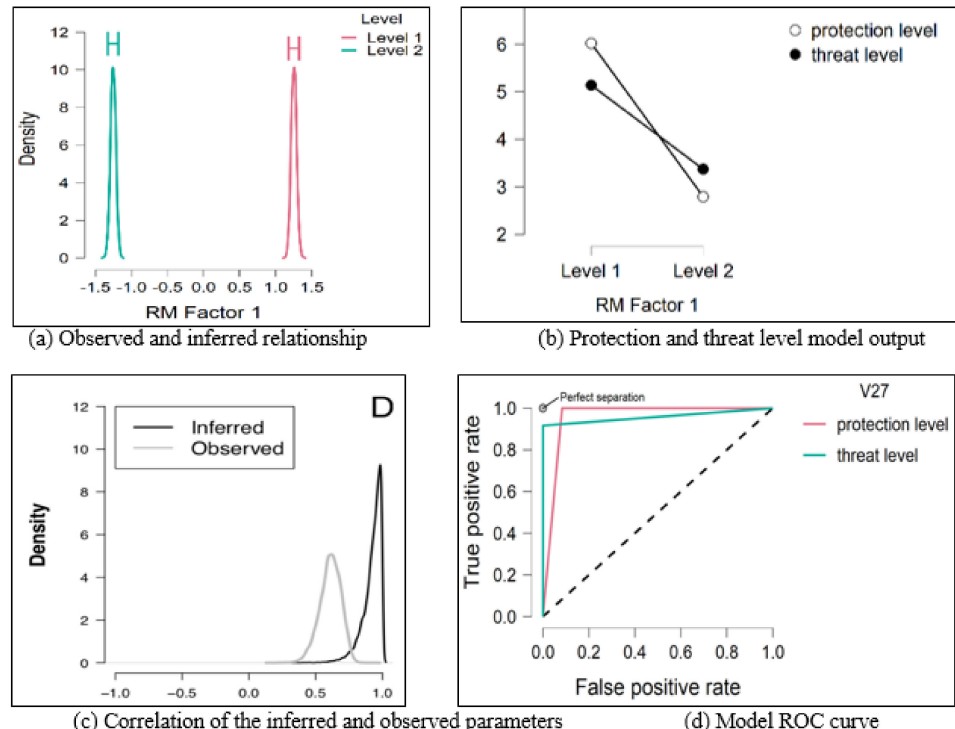

(a) Observed and inferred relationship

(b) Protection and threat level model output

(c) Correlation of the inferred and observed parameters

(d) Model ROC curve

**Figure 17.** Model posterior distribution showing the correlation between the observed data (protection level) and the inferred data (threat level). Figure 17a shows the observed and inferred relationship of the RM factor and the density levels as detailed in Table 6. Figure 17b shows the protection and threat level correlation model output which demonstrates how the inferred and observed parameters in Figure 17c are related with reference to the ROC curve (Figure 17d).

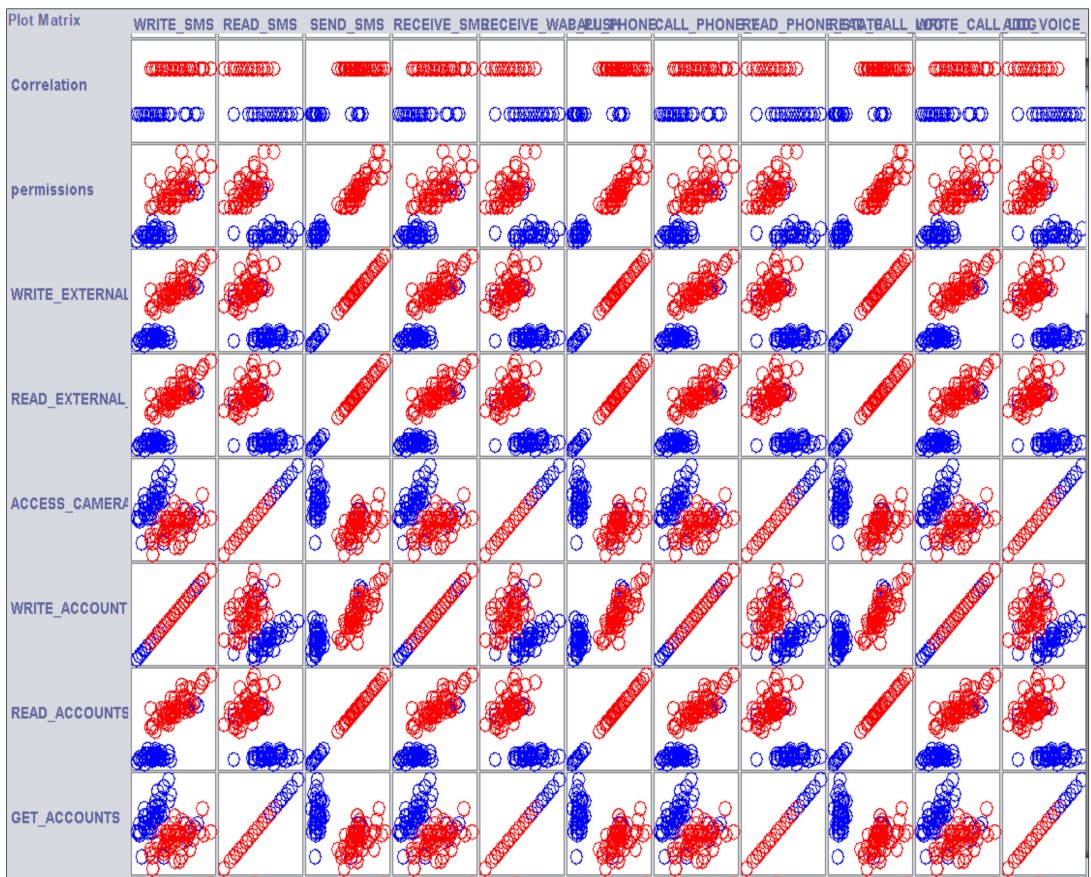

**Figure 18.** Plot Matrix showing exploratory plane analysis visualization of the correlation between threat and protection level in permissions. The red color indicates the threat level, while the blue indicates the protection level which forms the guarded region.

To segregate individual permissions and visualize their threat and protection level, we adopted a minimization approach [98]. Consider two permission parameters $\varphi$ and $\partial$ represent two different random variables. Let $\varphi$ represent the observed random variable and $\partial$ the unobserved, respectively. Let the minimization factor in the observed random variable be $\varphi_\varepsilon$ and $\partial_\varepsilon$, respectively. The separate factor between $\varphi$ and $\partial$ parameters is given as:

$$C\partial = \varphi + \varphi_\varepsilon \tag{27}$$

$$\varphi = \partial + \partial_\varepsilon \tag{28}$$

Figure 19 shows the sum of weights of individual permissions in relation to their protection and threat level.

To minimize error in the separation factor, we assumed that the minimization factor in both the observed and unobserved random parameters are distributed normally with their respective variances. Let the variances in the normal distribution for $\varphi_\varepsilon$ and $\partial_\varepsilon$ be $\sigma^2_{\varphi_\varepsilon}$ and $\sigma^2_{\partial_\varepsilon}$. The correlation that exists between $\varphi$ and $\partial$ with their given variances is minimized and expressed as:

$$\rho = \frac{\sigma\varphi\partial}{\sqrt{\sigma^2\partial\sigma^{2\varphi}}} \tag{29}$$

Using Behseta Bayesian correlation principles [99], error minimization in $\rho$ is expressed as follows:

$$\rho = \frac{\sigma\varphi\partial}{\sqrt{\left(\sigma^2_\varphi + \sigma^2_{\varphi_\varepsilon}\right) + \left(\sigma^2_\partial + \sigma^2_{\partial_\varepsilon}\right)}} \tag{30}$$

However, if an error in the measurement is not correlated with the observed and unobserved random parameters, then the correlation of the unobserved parameter is greater than that of the observed random parameters accordingly.

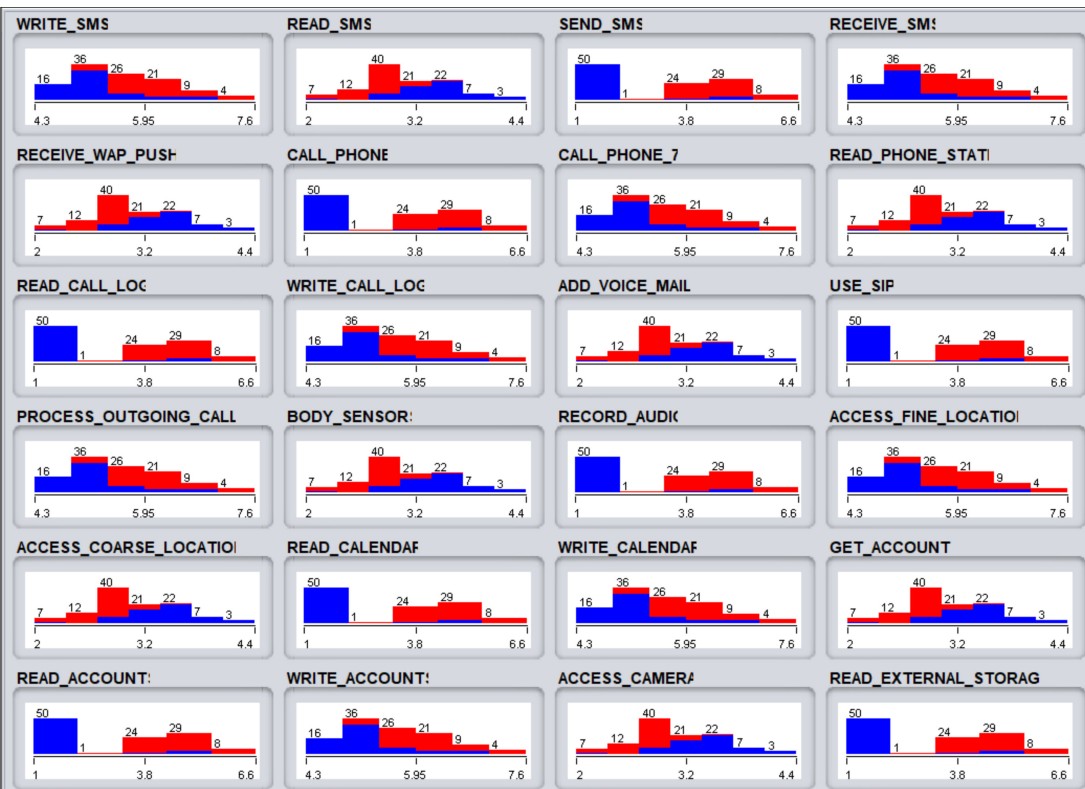

**Figure 19.** Exploratory factor plane visualization of permissions threat and protection level class correlation.

## 5. Results and Discussion

This section of the paper discusses some of the results obtained from the model and their implications. The research identifies the relationship between the protection and the threat level permission could present. In presenting the result, we selected a few permissions to test their threat and protection level relationship based on our model formulation. Using the correlation tables, the rest of the permissions can be tested by any researcher, mobile security experts, and Android users to determine the correlation that exists between the threat and protection level of each permission request. The Boxplots (Figure 20) and the distribution plots (Figure 21) for some permissions and their threat and protection levels relationship. We observed that the threat level is higher than the protection level in the ACCESS_CAMERA and READ_PHONE_STATE, respectively. On the other hand, the protection level in the RECORD_AUDIO and WRITE_EXTERNAL_STORAGE is higher than the threat level posed by those permission variables. Figure 22 shows the threat level of sensitive API extracted by our methodology.

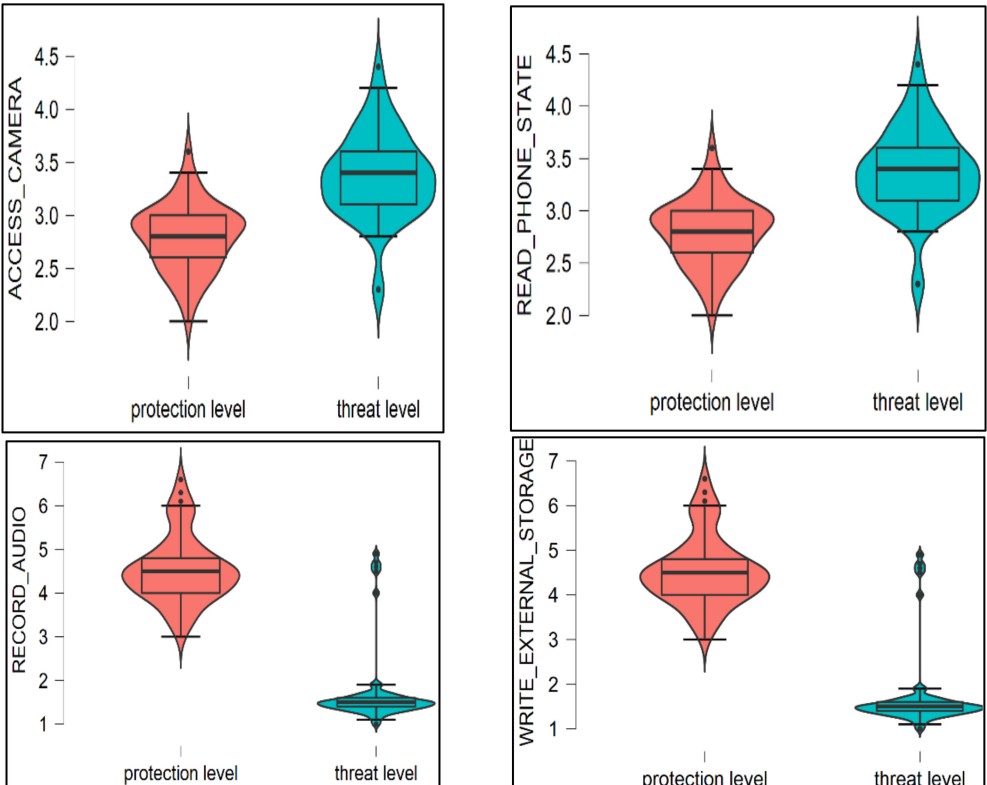

**Figure 20.** Boxplots showing correlation between protection and threat level of some selected dangerous permission variables.

We observed that each permission has a threat and protection level. Android permissions with the same protection level have the same threat level Figure 23, especially if they are in the same. Permissions in the same clusters have similar densities and attributes. We observe that permissions that are in the same cluster may exhibit similar attributes and if coming from a malicious application, such permissions behave similarly in their effects (Appendix A shows details of how the cluster tree was formed using a decision tree algorithm). This implies that if any of them request access, the other ones in the same set will tend to execute some functionalities in the background.

The research identifies that permissions are interposed in sets and can only be associated with a set with higher density. According to the results produced by the model, Bayes factors that are one-sided signify substantiation for the non-existence of correlation between threat level and protection level for both the inferred and observed relationship. However, this is not the situation in Android applications. As shown in Figure 23, the result shows that Android malicious applications have different threat levels. Under the risk heading, the threat level posed by an application is ranked from 1 to 10 using rectangular dots. If the risk level of an application to a device is more than 6, our model classifies the permission as dangerous based on the protection level of the permission name. Conversely, if permission has a protection level whose threat level is less than 7, our method classifies such permission as dangerous.

From the result obtained, we can infer that if a malicious application has a low threat level, it is highly unlikely for such an application to dangerously infect a mobile device. At a moderate threat level, an attack on a device is potential but not likely. However, in substantial threat level applications, attack on mobile devices is likely and highly likely in severe threat levels (ranked up to 7 in our model). Permissions requested by applications with severe threat levels are likely to be dangerous permissions, as illustrated in Figure 24. Applications with critical threat level (ranked from 8 to 10 in our model) are highly likely to infect a mobile application and could request a combination of both normal and dangerous permissions.

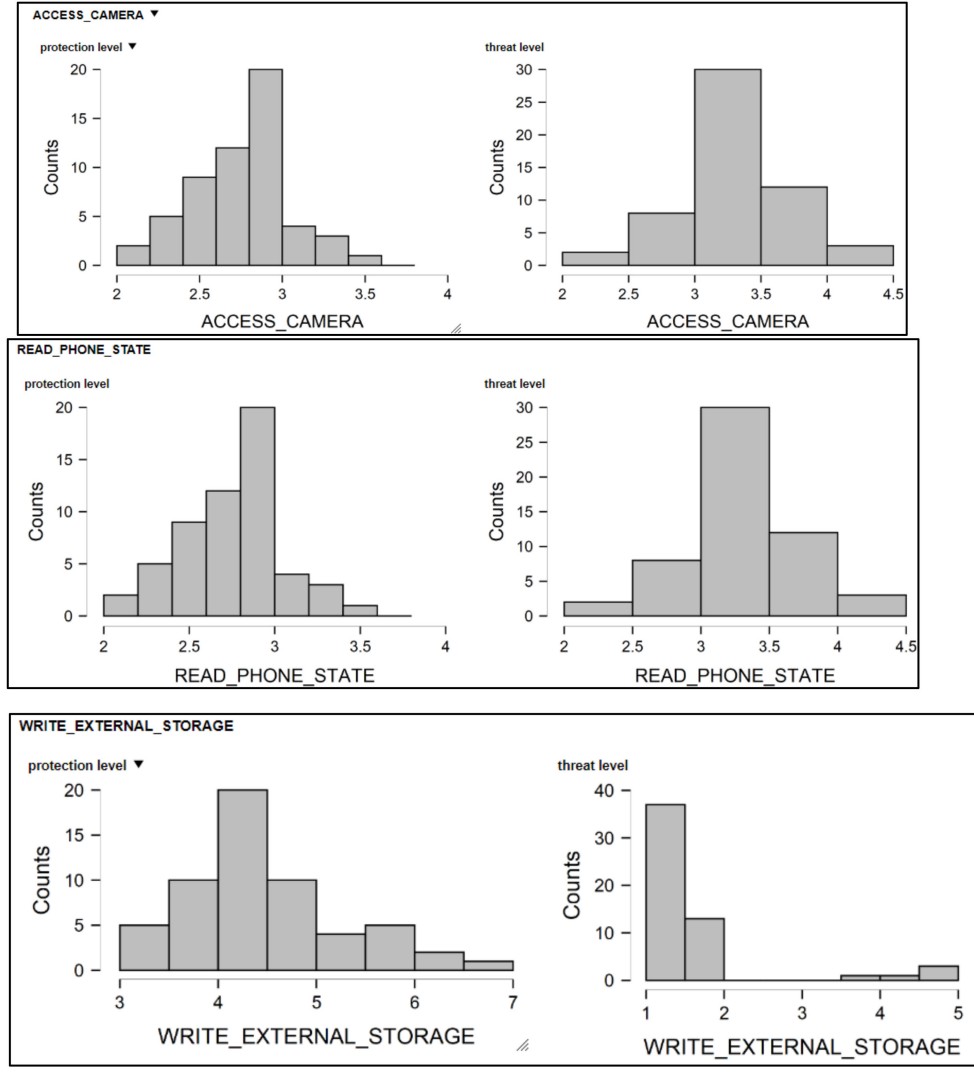

**Figure 21.** Scatter plots showing the protection and threat level of each permission per count.

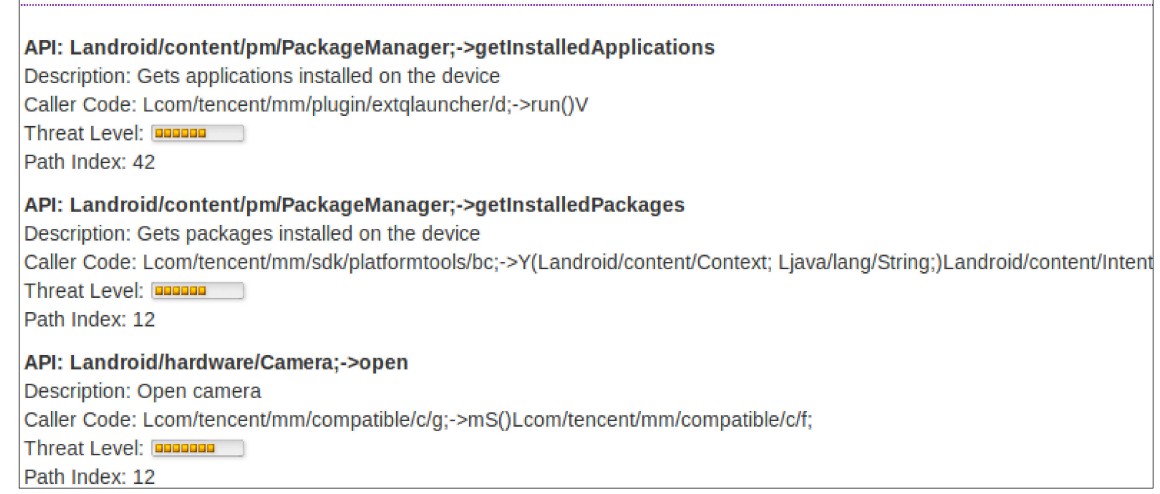

**Figure 22.** Visualization of some sensitive API showing their path index, threat level, caller code, and operation.

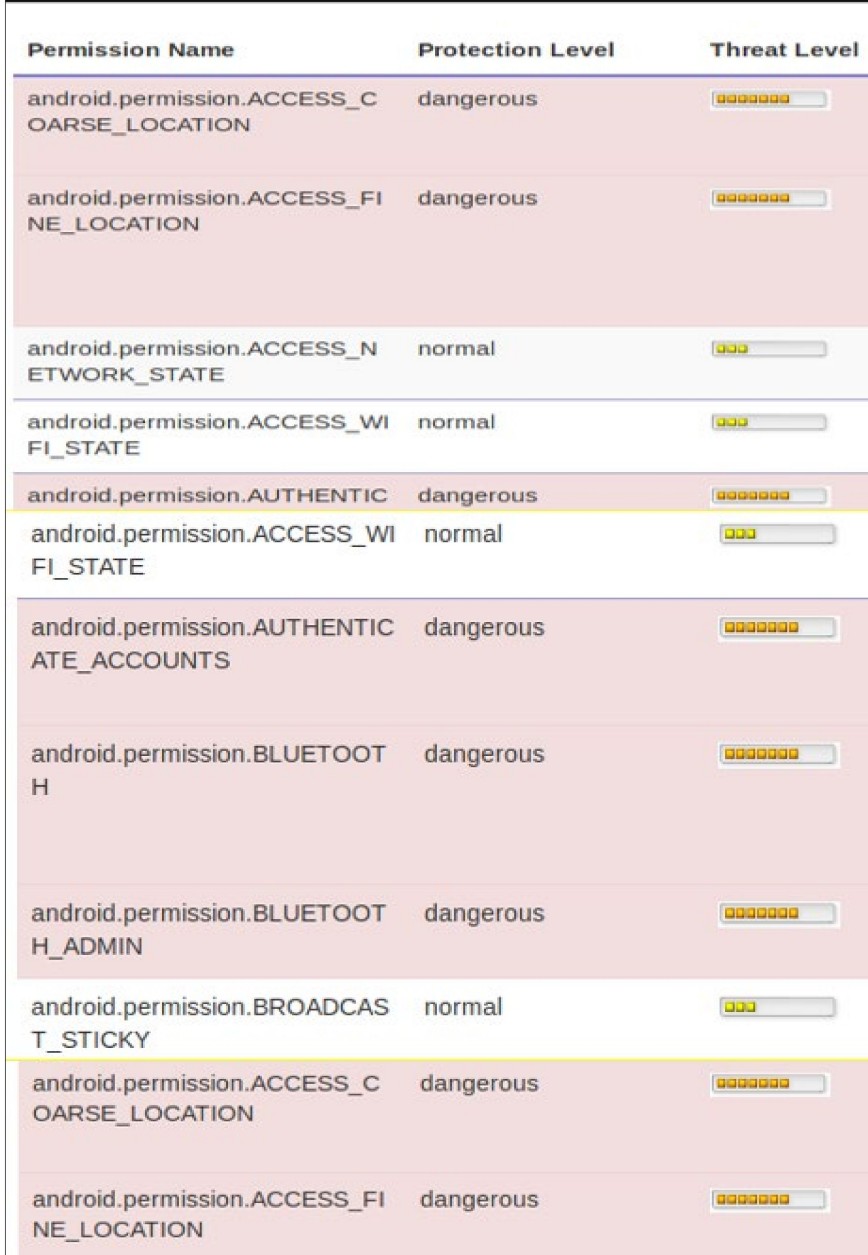

**Figure 23.** Classifying protection and threat level of certain permissions based on their usage.

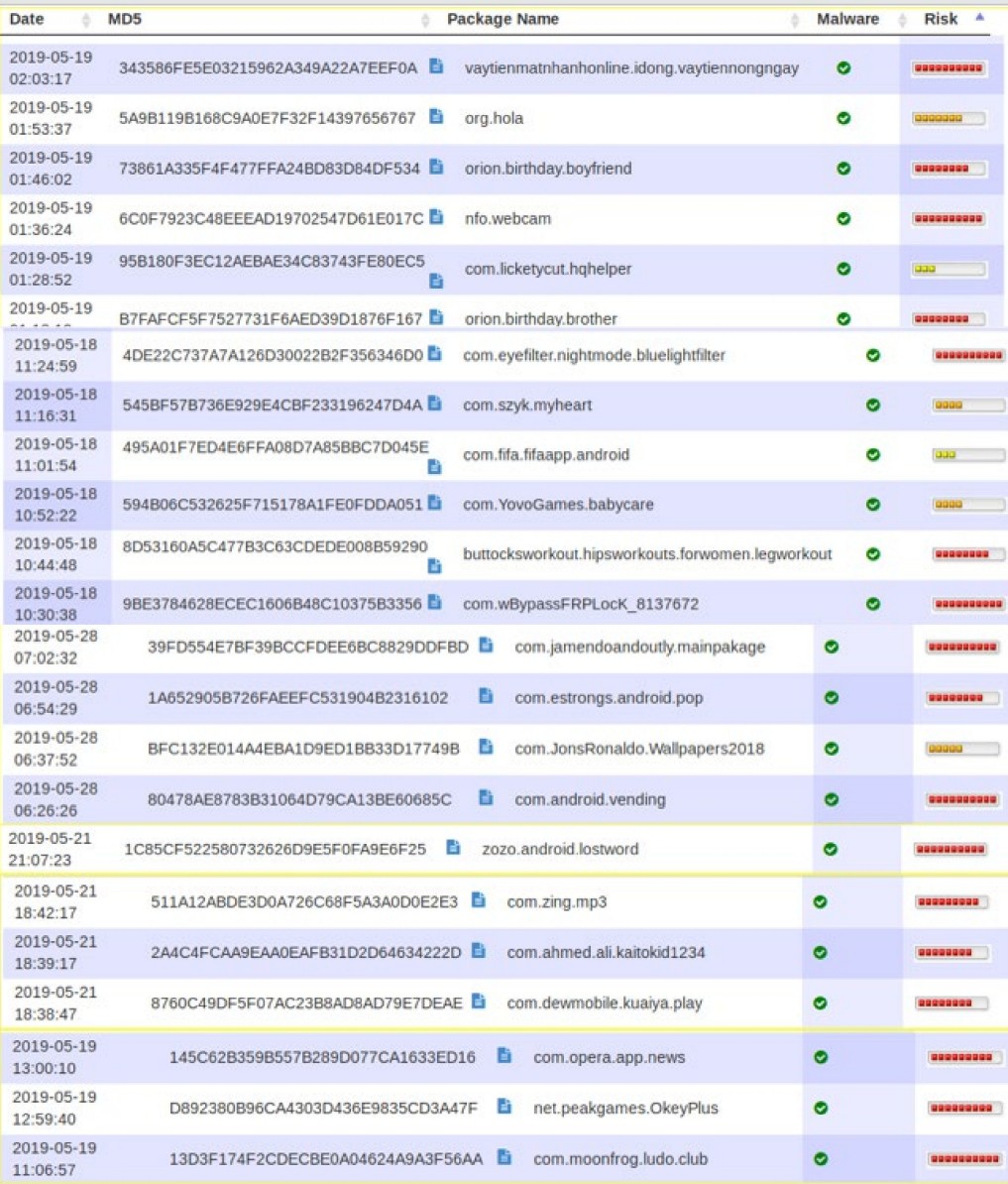

**Figure 24.** Android applications threat level classification.

During clustering analysis, the results obtained (as shown in Figure 25) demonstrated that most permissions in the same cluster have similarities in the feature distribution. For instance, READ_CALL_LOG and RECORD_AUDIO have some similarities in their structural distribution as can be seen in RECEIVE_SMS and WRITE_SMS respectively. This also demonstrated that permission in the same cluster have relevant features that are correlated.

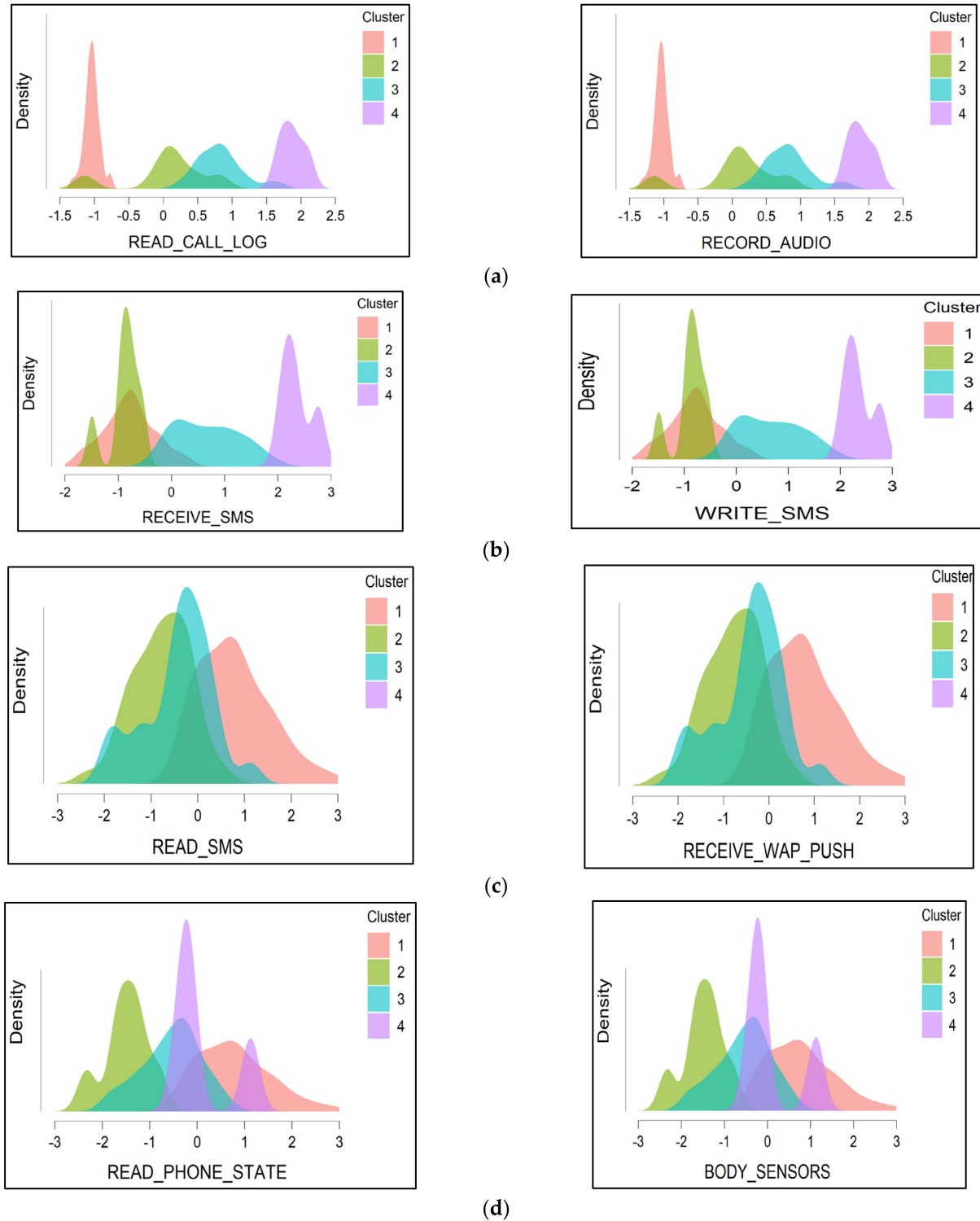

**Figure 25.** Cluster density plots showing dangerous permission-based clustering. (**a**) Permissions in cluster 1. (**b**) Permissions in cluster 2. (**c**) Permissions in cluster 3. (**d**) Permissions in cluster 4.

We observe that correlated permissions usually appear in locations that are the same. This suggests that Android applications that request CALL_LOG, SMS, or any permission sets that are correlated, similar permission sets will be requested underground and will be operational in the same region. Permissions in the same region execute frequent tasks such as read SMS and access audio and camera, among others. However, not all frequent permissions are dangerous. For example, a permission request to access the internet does

not invoke internet access control obtained by an application. We then inferred that internet permission does not support adequately grained management or control of the device resources. Comparatively, rather than use direct permission features to gain control of a device, malware authors may define some functionalities as access control. This aids in specifying how the malicious program can communicate with pre-defined applications to access the resources in question without necessarily requesting permission related to such resources. Based on the results obtained, we recommend improvement in the permission model of Android OS by differentiating internet permission access and other similar permissions into distinct groups and classes. Let a clear mechanism be provided for specifying Android permissions that are correlated with device resources. When the access control list is specified, this will help permissions definition by developers without using permissions attributes that are self-defined. For Android, numerous permissions have effects on Android end users and even developers. It is then pertinent to configure a system to help understand each permissions' action concerning the device resources rather than over-requesting them by the developers simply because they want an app to function. This could be a good security strategy in preventing a malware infection on mobile devices in general. However, studying actions associated with each permission is beyond the scope of our research.

*Attackers and Permission Request to Escalate Its Privileges*

This subsection provides a short description of how an attack leveraged misplaced trust in a permission request to escalate its privileges as illustrated in Figure 26. Malware, like any other program, can potentially execute any permission from a standard user to root (administrator) based on the context it was originally executed within. However, an Android application version that has less permission is not restricted to access components of a more privileged application. For instance, consider X, Y, and Z represent three different Android applications, each running in its own sandbox and having two components. In application Z, component $Z_{z1}$ is protected by permission p1. Similarly, $Z_{z2}$ by p2. Application Y is granted permission p1; hence, $YZ_1$ and $YZ_2$ can access $Z_{z1.}$ However, Y is not protected by any permission, and its components are publicly accessible. X does not have any permissions, but its component $Z_{x1}$ can access $Z_{x1}$. $Z_{x1}$ is not accessible to $Z_{z1}$ directly since it does not have permission p1; however, it can do so through component $X_{Y1}$. This is a privilege escalation attack since the privileges of application X are escalated by the attacker to the privileges of application Z indirectly through Y, which indicates transitive usage of permission privileges. To prevent this kind of permission escalation attack, application Y must ensure that another application calling it must have permission p1.

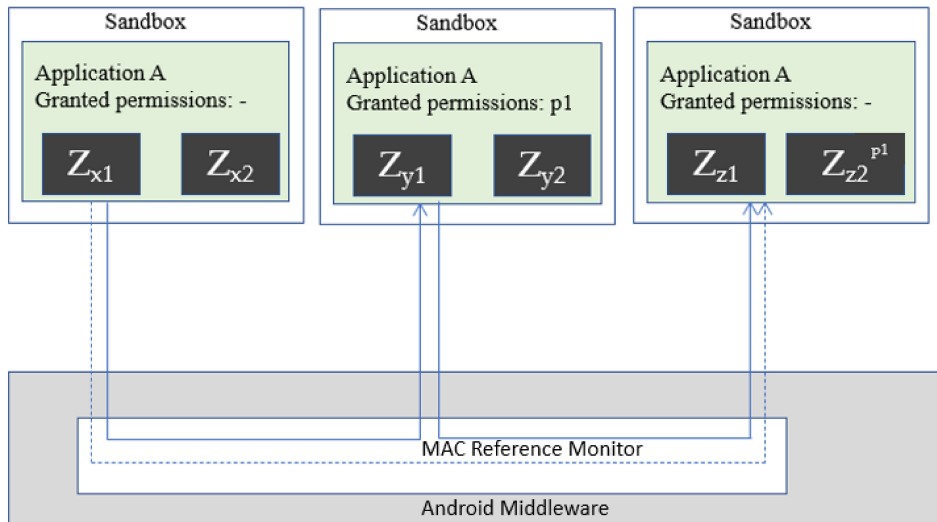

**Figure 26.** Demonstrating how an attack leveraged misplaced trust in a permission request to escalate its privileges.

## 6. Conclusions

In this work, we carried out factor analysis to determine the correlation between dangerous permission variables for Android malware samples using Bayesian correlation. The aim is to establish if there is a correlation in the set of a given permission request variables. This method is to assist in the analysis of the Android permissions security structure. Using SOM and t-SNE techniques, we visualized the Android malware data set by applying exploratory factor plane analysis, which reviewed patterns in which permissions are related. Large data set consisting of 12,267 malicious and 10,837 benign applications with different categories were used for the experiments.

Our results demonstrate our model can correctly recognize the correlation between dangerous permissions. We identified that permissions that are in the same protection level have the same threat level. Visualization results indicate that most permissions in the large subcategory were requested by very few Android applications, while frequently used permissions were those in the small subcategory. This signifies that there is no adequate expressiveness from permissions that are requested frequently. We infer that those not frequently requested could be disintegrated into the common class. As a result, we suggest adding finer granularity as a security approach for Android permissions that are frequently requested by applications will enhance this expressiveness and enhance Android security, especially when combined with the ones occasionally requested.

In general, this method will facilitate researchers infer correlation in the presence of estimation uncertainties in distributed clusters of Android permissions. The understanding of high threat permission will assist in enforcing security parameters at the permission security system of the Android platform and other mobile OS. These findings are significant for offering insights to assist mobile users in making low-risk decisions during application installations and when granting access to applications that have high threat levels, especially when similar permission names with the same protection level are requested. This provides insight to researchers and android users to infer the latent relationship between (1) Android permission requests from malicious Android applications; (2) the threat level and protection level of a permission request; and (3) to understand how a single grant of one permission may trigger background actions of others that are correlated with similar attributes. Although our model used three correlation coefficients, further research can be conducted on using more correlation coefficient parameters to model the relationship between permissions based on their threat and protection level. We used only the Android malware data set for this model; we suggest that malware data set from other mobile platforms could be used as a further study to explore their permissions architecture using our methodology.

In conclusion, with the number of malicious applications increasing daily at a very fast rate, vulnerabilities are also increasing, making the platform for attackers wider. Thus, there is a need for a technique that provides a complete solution against permission privilege escalation attacks along with satisfying all the usability requirements. In the future, the impact on the security of Android users and vendors using Android permissions at different protection levels might be studied vigorously. Other permission systems on other mobile operating systems could be investigated, such as Windows and iOS. Moreover, further studies could investigate more if the defined permissions of the existing applications are fully utilized by them, or they are classified as over-privileged applications.

**Author Contributions:** Conceptualization, methodology, software, M.A.; validation, S.M.; original draft preparation, M.A.; writing—review and editing M.A. and S.M.; supervision, S.M.; project administration, S.M.; funding acquisition, M.A. All authors have read and agreed to the published version of the manuscript.

**Funding:** This research was funded by the Petroleum Technology Development Fund (PTDF), grant number PTDF/ED/PHD/AMA/1245/17/17.

**Data Availability Statement:** The data presented in this study are available on request from the corresponding author. The data are not publicly available due to the nature of the dataset [malicious files].

**Conflicts of Interest:** The authors declare no conflict of interest.

## Appendix A. Decision Cluster Tree

*Tree*

```
    android.permission.DISABLE_KEYGUARD > 0.500
    | android.permission.SEND_SMS > 0.500
    | | android.permission.CALL_PHONE > 0.500:  cluster_0 {cluster_0=33, cluster_1=0, cluster_2=0,
cluster_3=0}
    | | android.permission.CALL_PHONE ≤ 0.500:  cluster_1 {cluster_0=0, cluster_1=1, cluster_2=0,
cluster_3=0}
    | android.permission.SEND_SMS ≤ 0.500
    | | android.permission.ACCESS_WIFI_STATE > 0.500:  cluster_3 {cluster_0=0, cluster_1=0,
cluster_2=0, cluster_3=3}
    | | android.permission.ACCESS_WIFI_STATE ≤ 0.500:  cluster_2 {cluster_0=0, cluster_1=0,
cluster_2=3, cluster_3=0}
    android.permission.DISABLE_KEYGUARD ≤ 0.500
    | android.permission.READ_PHONE_STATE > 0.500
    | | android.permission.INTERNET > 0.500
    | | | android.permission.RECEIVE_SMS > 0.500
    | | | | android.permission.WRITE_SMS > 0.500
    | | | | | type > 0.500:  cluster_1 {cluster_0=0, cluster_1=18, cluster_2=0, cluster_3=0}
    | | | | | type ≤ 0.500:  cluster_3 {cluster_0=0, cluster_1=0, cluster_2=0, cluster_3=1}
    | | | | android.permission.WRITE_SMS ≤ 0.500
    | | | | | android.permission.READ_CONTACTS > 0.500
    | | | | | | android.permission.CALL_PHONE > 0.500:  cluster_3 {cluster_0=0, cluster_1=0,
cluster_2=0, cluster_3=1}
    | | | | | | android.permission.CALL_PHONE ≤ 0.500:  cluster_1 {cluster_0=0, cluster_1=1,
cluster_2=0, cluster_3=0}
    | | | | | android.permission.READ_CONTACTS ≤ 0.500:  cluster_3 {cluster_0=0, cluster_1=0,
cluster_2=0, cluster_3=12}
    | | | android.permission.RECEIVE_SMS ≤ 0.500
    | | | | type > 0.500
    | | | | | android.permission.ACCESS_NETWORK_STATE > 0.500
    | | | | | | android.permission.ACCESS_WIFI_STATE > 0.500:  cluster_3 {cluster_0=0, cluster_1=0,
cluster_2=0, cluster_3=92}
    | | | | | | android.permission.ACCESS_WIFI_STATE ≤ 0.500
    | | | | | | | android.permission.READ_CONTACTS > 0.500:  cluster_3 {cluster_0=0, cluster_1=0,
cluster_2=0, cluster_3=8}
    | | | | | | | android.permission.READ_CONTACTS ≤ 0.500
    | | | | | | | | android.permission.VIBRATE > 0.500:  cluster_2 {cluster_0=0, cluster_1=0,
cluster_2=1, cluster_3=0}
    | | | | | | | | android.permission.VIBRATE ≤ 0.500
    | | | | | | | | | android.permission.GET_TASKS > 0.500:  cluster_3 {cluster_0=0, cluster_1=0,
cluster_2=0, cluster_3=2}
    | | | | | | | | | android.permission.GET_TASKS ≤ 0.500
    | | | | | | | | | | android.permission.RECEIVE_BOOT_COMPLETED > 0.500:  cluster_3
{cluster_0=0, cluster_1=0, cluster_2=0, cluster_3=2}
    | | | | | | | | | | android.permission.RECEIVE_BOOT_COMPLETED ≤ 0.500:  cluster_2
{cluster_0=0, cluster_1=0, cluster_2=1, cluster_3=0}
    | | | | | android.permission.ACCESS_NETWORK_STATE ≤ 0.500
```

```
    | | | | | | | android.permission.ACCESS_COARSE_LOCATION > 0.500:  cluster_3 {cluster_0=0,
cluster_1=0, cluster_2=0, cluster_3=9}
    | | | | | | | android.permission.ACCESS_COARSE_LOCATION ≤ 0.500
    | | | | | | | | android.permission.CHANGE_WIFI_STATE > 0.500:  cluster_3 {cluster_0=0,
cluster_1=0,
cluster_2=0, cluster_3=2}
    | | | | | | | | android.permission.CHANGE_WIFI_STATE ≤ 0.500
    | | | | | | | | | android.permission.ACCESS_FINE_LOCATION > 0.500:  cluster_3 {cluster_0=0,
cluster_1=0, cluster_2=0, cluster_3=1}
    | | | | | | | | | android.permission.ACCESS_FINE_LOCATION ≤ 0.500:  cluster_2 {cluster_0=0,
cluster_1=0, cluster_2=6, cluster_3=0}
    | | | | type ≤ 0.500
    | | | | | android.permission.ACCESS_COARSE_LOCATION > 0.500:  cluster_3 {cluster_0=0,
cluster_1=0, cluster_2=0, cluster_3=3}
    | | | | | android.permission.ACCESS_COARSE_LOCATION ≤ 0.500
    | | | | | | android.permission.CHANGE_WIFI_STATE > 0.500:  cluster_3 {cluster_0=0, cluster_1=0,
cluster_2=0, cluster_3=1}
    | | | | | | android.permission.CHANGE_WIFI_STATE ≤ 0.500:  cluster_2 {cluster_0=0, cluster_1=0,
cluster_2=8, cluster_3=0}
    | | android.permission.INTERNET ≤ 0.500:  cluster_2 {cluster_0=0, cluster_1=0, cluster_2=5,
cluster_3=0}
    | android.permission.READ_PHONE_STATE ≤ 0.500
    | | android.permission.CHANGE_WIFI_STATE > 0.500
    | | | android.permission.ACCESS_COARSE_LOCATION > 0.500:  cluster_3 {cluster_0=0, cluster_1=0,
cluster_2=0, cluster_3=2}
    | | | android.permission.ACCESS_COARSE_LOCATION ≤ 0.500
    | | | | android.permission.ACCESS_NETWORK_STATE > 0.500:  cluster_3 {cluster_0=0, cluster_1=0,
cluster_2=0, cluster_3=1}
    | | | | android.permission.ACCESS_NETWORK_STATE ≤ 0.500
    | | | | | android.permission.READ_CONTACTS > 0.500:  cluster_3 {cluster_0=0, cluster_1=0,
cluster_2=0, cluster_3=1}
    | | | | | android.permission.READ_CONTACTS ≤ 0.500:  cluster_2 {cluster_0=0, cluster_1=0,
cluster_2=4, cluster_3=0}
    | | android.permission.CHANGE_WIFI_STATE ≤ 0.500
    | | | android.permission.ACCESS_COARSE_LOCATION > 0.500
    | | | | android.permission.ACCESS_WIFI_STATE > 0.500:  cluster_3 {cluster_0=0, cluster_1=0,
cluster_2=0, cluster_3=2}
    | | | | android.permission.ACCESS_WIFI_STATE ≤ 0.500:  cluster_2 {cluster_0=0, cluster_1=0,
cluster_2=5, cluster_3=0}
    | | | android.permission.ACCESS_COARSE_LOCATION ≤ 0.500:  cluster_2 {cluster_0=0, cluster_1=0,
cluster_2=169, cluster_3=0}
```

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
