# Peer review of "Modeling Correlation between Android Permissions Based on Threat and Protection Level Using Exploratory Factor Plane Analysis"

_jcp, doi:10.3390/jcp1040035_

Round 1

Reviewer 1 Report

The research identifies the relationship between the protection and the threat level permission could present. Using the correlation tables, the rest of the permissions can be tested by any researcher, mobile security experts, and Android users to determine the correlation that exists between the threat and protection level of each permission request.

The paper is well written and has scientific validity. Some issues that can significantly improve its quality are presented below:

  1. A detailed description must be included in the paper that emphasizes the main pros and cons of the authors’ proposal with regard to the state of the art.
  2. The authors must extend the explanation of how the proposed technique can be extended to cover a wider scientific area without reducing the main points that are currently described.
  3. I think that if the authors wish this paper is well considered by experts in the machine vision and machine learning communities, more attention should be devoted to discussing the application scenario. I suggest simplifying it or better explain with realistic examples.
  4. There are many up-to-date theoretical studies on Machine Learning and well-established communities working on different theoretical aspects and techniques (e.g. make your network shallower by fewer layers, use less number of hidden units, decrease regularization, etc, for example, https://link.springer.com/chapter/10.1007/978-3-319-27478-2_20, https://ieeexplore.ieee.org/document/8888430, https://ieeexplore.ieee.org/document/9065765). The authors must extend the explanation about the main differences between the current submission and the previous studies. I would suggest a comparison study.
  5. The authors should probably provide more information about the proposed architecture. This is a major issue of the paper of how the authors have chosen this specific architecture for the proposed processing method, how it emerged and why the proposed architecture is the optimal solution.
  6. The second major issue of the paper is the explanation of the results, which are presented casually and without thorough analysis.
  7. Figures are small and apparently of low resolution. If the authors consider that provides important information, they should definitely enlarge it to be clear and legible.
  8. The discussion section could include some contributions to the international literature.
  9. Finally, the authors do not include a detailed description of how the method proposed can be extended. This is a serious omission in the paper, as it indicates the evolution and progress failure of the proposed method and of the scientific field in which the authors are specified.

Author Response

Thank you very much for taking the time to review our manuscript to improve its quality. We appreciate it gratefully. Apart from the comments addressed, both authors worked together to improve the research background, research design, method description and results in the presentation of the paper.  Please, below are the efforts we have made in responding to your review comments:

1 and 2

Thank you very much for the valid comment. The authors have added paragraphs in section 4.4, and sentences in section 5 and 6 respectively regard to the state of the art and the need for need for a technique that provides a complete solution against permission privilege escalation attacks along with satisfying all the usability requirements as highlighted in the manuscript accordingly. Also, the authors as highlighted in the manuscript gave some explanation of how the proposed technique can be extended to cover a wider scientific area 

3 and 4

Thank you very much for the valid comment. As observed by the reviewer, the authors added a subsection 5.1 to discuss with realistic illustrations and examples how in real scenario, an attacker can escalate permission privileges to infect and Android device discussing the application scenario. The authors in their previous published paper (https://doi.org/10.1002/spy2.164)  which is part of this research discussed regularization extensively. The authors believe that since this very manuscript is a built up of the previous published work, we made reference to the paper for so of those information to avoid repetitions in the publications.

5

Thank you very much for the valid comment. In section 4.0 which is the material and method, the authors added a paragraph which provided more information about the proposed architecture. However, because the authors used the same malware data set and the same methodology as the one the used in their previous work  (https://doi.org/10.1002/spy2.164) which this very paper is a continuation, the authors avoided providing elaborate information on this to help make the paper concise and avoid repetition. However, the authors referenced the previous work as highlighted in section 4.

6 and 7

Thank you very much for the valid comment. To improve the analysis of the results, the authors added some sentences, paragraphs, and sections in the correction copy as highlighted in the revised copy of the manuscript. Apart from result analysis, the authors also made considerable changes on the results and the general manuscript. Authors were checked and, in most cases, replaced with those of higher resolutions as generated by the models in each instance. For instance, the authors could not generate a better resolution from the model. However, we increased the size of the figures to make it clearer and more readable.

8 and 9

Thank you very much for the valid comment. To include contribution to the international literature, subsections and paragraphs were added by the authors according as validly observed by the reviewer. for instance, we have added subsection 5.1 to describe the connection between the results and how permissions can be escalated by an attacker as highlighted in the section accordingly. The authors added a section that discussed the implication of the result on how attackers use permission request to escalate privileges. The authors added a paragraph including a detailed description of how the method proposed can be extended.

Reviewer 2 Report

Reading the introduction, I found this paper very promising, but it did not stand up to expectations.

Starting from section 2.2, language becomes more imprecise and confusing: 

2.2.1

"users do not need the approval to grant access"?

And the subsequent sentence looks contradicting or under-explained.

2.2.2

what is "the certificate of declaring and requesting application"?

"If it can be signed"?

What "condition" are authors referring to?

2.3

"intents create unique and up-to-date instances" of what?

"calling a simple operation"? 

What is an "explicit intent"?

"By doing so" sentence is written twice, identical. 

The whole subsection gives the impression of stating the obvious, often repeating generic sentences in the tone of "intents play a significant role" without really explaining how and why.

2.4

defines API acronym after it has been used many times. Give definitions at first usage.

APIs "reinforce third-party applications"?

"is easily is made easy"

APIs "occupy the SDK"?

3.

repetitions:  "following categories  as follows", "accuracy of 81,5% accuracy"

"many other effects"?

4.1

repeats "previous" three times in 4 lines 

4.4

"task demanding" -> "demanding task" (not the only case of adjective inversion, look for others)

" a total of 158 permissions" - must clarify the meaning of this number

"serum dataset"? serum is only used in biology.

4.5

end of page 17. "The summary version." is not a sentence.

5. 

Third row of figure 21 is badly drawn

Page 28, last line of text is not a sentence, but a copy the caption of fig. 22

End of page 30 "we can only be associated"... we?

Second sentence at page 33: do you mean "WITHIN Android applications"? Sentence starting with "Thus" has no verb. 

Many wrong plurals are strewn across the paper.

Enough with clarity, there are many more examples but my job is not to proofread. Contents discussion now.

2.2.2 

Authors state this is the highest privilege but I argue 2.2.4 is higher.

2.2.4 Halfway through the subsection, a general topic is introduced, not specific of the signatureOrSystem permission. Moreover, this paragraph (starting at "Each Android protection...") suddenly introduces many technical details, which are not understandable without prior definition.

2.4

"let's consider the case of an application that wants do get the identity of a device" is followed by a much more general case, not the stated one.

Side note: there are many highlighted sections, I suppose this is a revised version of an original manuscript.

The first of these, in 2.4 seem to drop out of the blue, it is not so smoothly connected to the preceding discussion, and in any case, I would not say that CFG can be attacked. CFG are a representation of possible code executions, if the code is changed the CFG changes too, but the attack is to code, not to its representation.

3. 

Near the end, a study is cited with high accuracy, noting the usage of 22 permissions as a limitation. I would say that it is a plus, in terms of efficiency.

The bulk of section 4 is an incredibly dense discussion of statistical methods, SOMs, and their usage in classification, showing two significant issues:

  • no effort to make it accessible to the non-expert reader
  • sparse, unfounded, unexplained connections to the paper's main subject

Before page 12, no hint is given on how these models are fed by permission data, yet there appears a SOM matrix citing them.

Statements such as "we apply equamax exploratory factor analysis rotation" or "using correlation methods [by] Pearson [et al]" may be technically correct, but the meaning of these sentences in context is left as an exercise to the reader.

Strong opinion:  readers of this paper are not willing to perform exercises, they want to be guided through analysis methods, to be convinced of their soundness for sure, but mainly to understand

  • why they are applicable to the specific topic of Android permission analysis
  • how permission features are converted into inputs for these methods

These two key aspects, I could not find them clearly explained anywhere.

Take page 17 as an example: In the words of the authors, they focus on the relationship between threat and protection, However, they spend two pages detailing the mathematical details of how to estimate the uncertainty of correlation and other parameters, instead of explaining why these are relevant. Every now and then, the names of some permissions appear, with no justification of why those and not others.

Section 5 promises a long-awaited discussion of results, but it is as shallow and vague as section 4 is overly detailed and technical (about the wrong topics). Most of the significance of the statistical analysis is explained too quickly, and recommendations take up 8 lines of text, while I expected a whole section devoted to them. 

Up and including to section 6, there is not a single concrete example of the relationship occurring between threat level and protection level, e.g. a short description of how an attack leveraged misplaced trust in a permission request to escalate its privileges. 

In short, it is an indisputable effort at applying sound scientific methods to a security problem; it is a real pity that it falls so short of conveying its meaning, of clarifying its formal links to the main subject, and of drawing detailed conclusions.

Author Response

REVIEWER 2

Thank you very much for taking the time to review our manuscript to improve its quality. We appreciate it gratefully. Apart from the comments addressed, both authors worked together to improve the research background, research design, method description and results in the presentation of the paper.  Please, below are the efforts we have made in responding to your review comments:

2.2.1

Thank you for the valid observation. However, we meant that Malware authors may define some functionalities as access control mechanisms to communicate with pre-defined applications to access the resources. The authors have removed the statement “Users do not need the approval to grant access to normal protection levels because the threats normal protection level presents are low and minimal” as observed by the reviewer.

2.2 2

Thank you for the valid observation. The authors meant that a signature-level permission means that the app defending itself with that permission (e.g., via android: permission attributes) and the app trying to talk to the first app that needs the permission ( element) must be signed by the same signing key. If the app defending itself is part of the device firmware, or is the OS itself, only apps signed by the same signing key as that firmware can talk to the defending app by holding the permission. The condition we refer to is that the certificate of the declaring application must be signed by the same key of the defending application.

The last sentence in the section was removed as accurately identified by the reviewer. Another sentence “; that is to say signature permissions can only be granted if the same certification signs both parties “was introduced to make the section complete. The authors also removed and modified statements that do not have a connection with the signatureOrSystem permission.

2.3

Thank you for the valid observation. The authors re-wrote the section by removing repeated sentences and presenting the most important information explaining how and why of the intent.

2.4

Thank you for the valid observation. The authors defined API acronym and corrected some of the mistakes in the section and many more. By the statement, “APIs occupy the SDK” the authors meant An API can be packaged in an SDK which is typically a set of software development tools that allows the creation of apps for a specific platform. Instead of “occupy”, the authors replaced the word with the package as highlighted in the manuscript. Many other errors identified in the section were corrected accordingly. Also, in section 3.0, the phrase “into the following categories” which was a repetition was removed. “as follow” was used. In section 3.1, the phrase “and many other effects” was a mistake. The authors wanted to write and many others or among others. As observed by the reviewer, “and many other effects” were removed from the statement since it is not significant. Other repetitions and errors observed were corrected also.

This section as highlighted is the revised version of the manuscript. The section was introduced because the second reviewer insisted that the authors should provide a paragraph on the control flow graph. However, as observed, the section was read through by both authors, and corrections were made accordingly. Inappropriate sentences were deleted such as “Thus, changing the representation of the features introduced by the malware.”; “Let's consider a case of an application that wants to get the identity of a device. The application has to leverage all the essential API call series of the mobile OS platform by calling all the execution paths from the control flow graph (CFG).”

3.

Thank you for the valid observation. Very valid observations on leaving some sentences such as "we apply equamax exploratory factor analysis rotation" or "using correlation methods [by] Pearson [et al]" as an exercise to the reader. The content of the paper is much, and the paper length is already long. As a result, the authors felt it might not be necessary explaining some ideas that are already published to make those sections as concise as possible and to minimise the paper length as possible.  Regarding the applicability and how the permission features were converted into inputs, we have published part of the research https://doi.org/10.1002/spy2.164 detailing on how permission features are used for the research were converted as binary inputs into the methods. It was the same inputs that were used. As a result, the authors did not want to explain the whole process again in this part of the paper.

4.1

Thank you for the valid observation. The authors noted your valid observations and corrected the repeated word “previous” which occurred so many times in the section.

4.4

Thank you for the valid observation. As observed by the reviewer, the authors replaced the phrase task demanding with the adjective “challenging” as highlighted in the manuscript. Also, since specifying the number of permissions extracted is not significant, the authors removed the sentence “, a total of 158 permissions requested access.”

4.5

As validly observed by the reviewer, the phrase “The summary version” in section 4.5 was removed accordingly.

5.

Thank you for the valid observation. The figures were captured from the result of the model and were not manually drawn by the authors. We captured it as it appeared in the model result. However, as observed by the reviewer, we tried and expanded it to look a bit tidy. The copy caption of fig 22 appearing in the last line of page 28 “Visualisation of some sensitive API showing their path index, threat level, caller code, and operation” was deleted from the text. The sentence “we can only be associated” was read through and corrected.

Based on the reviewer’s comment, we added a subsection in section 5 and provided a short a description of how an attack leveraged misplaced trust in a permission request to escalate its privileges. 

This subsection provides a short description of how an attack leveraged misplaced trust in a permission request to escalate its privileges. Malware like any other program, can potentially execute any permission from standard user to root (administrator) based on the context it was originally executed within. However, an Android application version that has less permission is not restricted to access components of a more privileged application. Forinstance, consider X, Y, and Z represent three different Android applications each running in its own sandbox and having two components. In application Z, component Zz1 is protected by permision p1. Similarly, Zz2 by p2. Application Y is granted permission p1, hence YZ1 and YZ2 can access Zz1. However, Y is not protected by any permission and its components are publicly accessible. X does not have any permissions, but its component Zx1 can access Zx1. Zx1 is not accessible to Zz1 directly since it does not have permission p1, however, it can do so through component XY1. This is a privilege escalation attack since the privileges of application X are escalated by the attacker to the privileges of application Z indirectly through Y which indicates transitive usage of permission privileges. To prevent this kind of permission escalation attack, application Y must ensure that another application calling it must have permission p1.

Round 2

Reviewer 1 Report

I have no further comments to make.

Author Response

Thank you very much for taking the time to review our manuscript again to improve its quality. The authors appreciate it gratefully.

Reviewer 2 Report

There are still language issues

2.2.2 Signature permissions ARE only... ;

2.3 signifant; ...

"During malware analysis, abstractions
which are FG targeting malware binaries to compare the rooted Trees and link the entry block of the application." ... botched syntax.

please have it checked by an editor, it is not my job to proofread).

I acknowledge the motivations given by the authors as a response to my previous review: "Regarding the applicability and how the permission features were converted into inputs, we have published part of the research https://doi.org/10.1002/spy2.164 detailing on how permission features are used for the research were converted as binary inputs into the methods. It was the same inputs that were used. As a result, the authors did not want to explain the whole process again in this part of the paper." 

However, I still believe

1) that a short summary of the process described in their previous paper should appear in this one. A reader can be asked to find details in another paper, not to read it to grasp the main meaning.

2) that the level of detail of section 4 is still disproportionate and hard to understand without specific knowledge of the used techniques. Even for a scientific paper, a layman explanation of what is the practical significance of the computed indicators would be appropriate.

Author Response

Dear reviewer 

RESPONSE TO COMMENTS

 Thank you very much for taking the time to review our manuscript again to improve its quality. The authors appreciate it gratefully. The authors with the help of an editor thoroughly went read the manuscript and corrected some errors in addition to the valid observations by the reviewer’s comments.  Please, below are the efforts we have made in responding to your review comments:

  • Thank you very much for your comment. In accordance with the reviewer’s comment, the authors in section 4.1, paragraph 2 provided a short summary of the process of how the feature were extracted from the data set and how they were converted to form input for the model as described in their previous paper. (To extract permissions from the selected Android applications, the Apktool was used to decompile the .apk file into different contents including ‘AndroidManifest.xml’, ‘Classes.dex’, and the ‘res’. All the applications permissions are contained in the ‘AndroidManifest.xml’ while the .dex has strings and Dalvik Opcodes respectively. The most signifcant features were selected using information gain by extracting the similarities between sets of permission and then calculating and scoring each permission individually. Using feature encoding, the permissions were converted into binary vectors by concatenating all the features, which is the input of the model. Details of applications and how permission features were extracted from the data set with other attributes and how they are converted into input vectors have been described in [72].
  • Thank you for the comment. The authors have made some significant changes in section 4 to enhance the understanding of non-technical persons in the field by changing some grammatical constructs and words. Some information that are not very significant in the section were removed by the authors to
  • Thank you very much for the comment. As highlighted in section 2.2.2 using a red tracker change, the word ‘are’ was added by the authors in line four of the section as rightly identified and recommended by the reviewer accordingly.
  • Thank you very much for the comment. In section 2.3, the authors located the spelling mistake in the word ‘signifant’ as identified by the reviewer and they corrected the mistake by changing the word to significant.
  • Thank you very much for the comment. In section 2.4, the authors carefully went through the whole paragraph and removed the sentence “During malware analysis, abstractions which are FG targeting malware binaries to compare the rooted Trees and link the entry block of the application” as identified by the reviewer. Also, the authors with the help of an editor made some changes and corrections to improve the quality of the manuscript. Some of the changes are highlighted below:
  • A comma (,) as highlighted  was inserted in the second to the last line of section 2.2.2 after the phrase that is to say,
  • In section 2.2.4 paragraph 2, we corrected the word resides and replaced it with reside in line 6 of the paragraph.
  • In section 2.3 line 4, the authors included ‘An’ before intent and ‘the’ before Android system in line 5 as highlighted in the main document. Also, the phrase perform operation was changed to ‘operate’. Also, the words ‘take’ and ‘view’ in line 8 and 9 were replaced with their present continues respectively.
  • In 2.4, ‘Android’ which was wrongly spelled was corrected as highted in line 2 of the section accordingly. The verb identifies at the beginning of line 2 of the same section was replaced with ‘identify’. In line 6, the article ‘an’ before Software Development Kits was removed and was replaced with the. In the second paragraph, the word ‘manipulates’ in line 6 was replaced with ‘manipulate’.
  • In section 4.0, the following words: ‘present, used, and understanding’ were corrected and used accordingly as highlighted in line 9, 11, and 12. Privileges, strengths, although, and other words wrongly spelled were also corrected as highlighted with a red marker in the manuscript respectively.
  • In section 4.4 paragraph 2, we added definite article in line 6.
  • In section 6.0 paragraph 4, errors in spelling and tenses were corrected as seen in lines 3, 5, and 7 respectively.